# Catchment tracers reveal discharge, recharge and sources of groundwater-borne pollutants in a novel lake modelling approach

Emil Kristensen[1], Mikkel Madsen-Østerbye[1], Philippe Massicotte[2], Ole Pedersen[1], Stiig Markager[2] and Theis Kragh[1]

[1]The Freshwater Biological Laboratory, Department of Biology, University of Copenhagen, Copenhagen, 2100, Denmark
[2]Department of Bioscience, Aarhus University, 4000, Roskilde, Denmark

*Correspondence to*: Emil Kristensen (emil.kristensen@bio.ku.com)

**Abstract**

Groundwater-borne contaminants such as nutrients, dissolved organic carbon (DOC), coloured dissolved organic matter (CDOM) and pesticides can have an impact the biological quality of lakes. The sources of pollutants can, however, be difficult to identify due to high heterogeneity in groundwater flow patterns. This study presents a novel approach for fast hydrological surveys of small groundwater-fed lakes using multiple groundwater-borne tracers. Water samples were collected from the lake and temporary groundwater wells, installed every 50 m within a distance of 5-45 m to the shore, were analysed for tracer concentrations of CDOM, DOC, total dissolved nitrogen (TDN, groundwater only), total nitrogen (TN, lake only), total dissolved phosphorus (TDP, groundwater only), total phosphorus (TP, lake only), $\delta^{18}O/\delta^{16}O$ isotope ratios and fluorescent dissolved organic matter (FDOM) components derived from parallel factor analysis (PARAFAC). The isolation of groundwater recharge areas was based on $\delta^{18}O$ measurements and areas with a high groundwater recharge rate were identified using a microbial influenced FDOM component. Groundwater discharge sites and the fractions of water delivered from the individual sites were isolated with the Community Assembly via Trait Selection model (CATS). The CATS model utilised tracer measurements of TDP, TDN, DOC and CDOM from the groundwater samples and related these to the tracer measurements of TN, TP, DOC and CDOM in the lake. A direct comparison between the lake and the inflowing groundwater was possible as degradation rates of the tracers in the lake were taken into account and related to a range of water retention times (WRTs) of the lake (0.25-3.5 years in 0.25 year increments). These estimations showed that WRTs above 2 years required a higher tracer concentration of inflowing water than found in any of the groundwater wells around the lake. From the estimations of inflowing tracer concentration, the CATS model isolated groundwater discharge sites located mainly in the eastern part of the lake with a single site in the southern part. Observations from the eastern part of the lake revealed an impermeable clay layer that promotes discharge during heavy precipitation events, which would otherwise be difficult to identify using traditional hydrological methods. In comparison to the lake concentrations, high tracer concentrations in the southern part showed that only a smaller fraction of water could originate from this area, thereby confirming the model results. A Euclidean cluster analysis of $\delta^{18}O$ isotopes identified recharge sites corresponding to areas adjacent to drainage channels, and a cluster analysis of the microbial influenced FDOM component C4 further identified five

sites, which showed a tendency towards high groundwater recharge rate. In conclusion, it was found that this methodology can be applied to smaller lakes within a short time frame providing useful information regarding the WRT of the lake and

more importantly the groundwater recharge and discharge sites around the lake. Thus, it is a tool for specific management of the catchment.

**Introduction**

Most lakes are connected to the groundwater, which to some degree defines their chemical and biological characteristics (Lewandowski et al., 2015). Particularly in smaller lakes and ponds, the groundwater contributes nutrients, dissolved organic

carbon (DOC), coloured dissolved organic matter (CDOM) or other contaminants, which can have a negative impact on the biological quality of the lakes (for nitrogen and phosphorous, see review by Lewandowski et al. (2015)). These inputs often result in unfavorable light conditions for submerged macrophytes due to either increased phytoplankton biomass (Smith, 2003) or increased light absorption from high CDOM concentrations. The negative impacts of the contaminants make the identification of pollutant sources an important management issue for lakes, which, however, is complicated for groundwater

due to temporal and spatial differences in discharge and associated pollutant concentrations (e.g., Meinikmann et al. 2013). In addition, the lake hydrology itself may be important, particularly in small water bodies. For example, low or fluctuating water level can have a large influence on the biodiversity of the lake (Chow-Fraser et al., 1998). This illustrates the need for approaches to quickly identify discharge (i.e. groundwater exfiltrating to the lake) and recharge (i.e. lake water infiltrating to the groundwater) areas at the lake-scale and thereby provide the necessary tools for an effective management strategy for

ponds and small lakes.

Groundwater discharge and recharge are traditionally identified and quantified by measurements of the hydraulic head through the installation of piezometers around the lake and of the hydraulic conductivity in the sediments (Rosenberry et al., 2015). This method is often combined with the use of seepage meters, which quantify the water entering or leaving through the lake bottom (Lee and Cherry, 1979). However, this method is challenged by the heterogeneous nature of groundwater

seepage related to the specific hydraulic conductivity of the lake bed sediments (Cherkauer and Nader, 1989; Kishel and Gerla, 2002; Rosenberry, 2005). Furthermore, these methods are also time-consuming as they have to be done several times throughout the season. The heterogeneity and annual variability in groundwater seepage call for a robust, easier and faster method to determine groundwater inputs and influences.

Various conservative tracers have been used to achieve estimates of groundwater flow and water retention times (WRTs) in

lakes. These tracers are divided into three main categories: (1) environmental tracers (natural derived tracers from the atmosphere or catchment which are transported to the system), (2) historical tracers (anthropogenic tracers such as 3H or 36Cl from nuclear testing) or (3) applied tracers (tracers added to the system such as Br, Cl or fluorescent dyes) (Stets et al., 2010). Precipitation-derived environmental tracers, such as the isotope $\delta 18O$ (reported in the Vienna-standard mean ocean water (SMOW) where $\delta sample$ ‰ = 1000((Rsample/Rsmow)-1) and R is the $\delta 18O/\delta 16O$ ratio (Turner et al., 1987)), have

been used to trace the interaction between ground- and surface water. As evaporation occurs in the surface water it becomes enriched with $\delta18O$ producing a unique lake $\delta18O/\delta16O$ ratio, which can be traced in the areas with groundwater recharge (Krabbenhoft et al., 1990). The isotopic composition can also be related to evaporation lines (from the local evaporation line describing $\delta18O$ and $\delta2H$ relationship) to estimate WRT (Gibson et al., 2002). Overall, these tracers provide information on flow patterns in the terrain or WRT, but they do not provide information regarding discharge in specific areas or the

concentrations of the previously mentioned pollutants in the discharging water. We propose a different approach utilising both conservative and non-conservative tracers such as dissolved carbon and nutrients, which are partly transferred to the percolating groundwater on its way to the lake (Kidmose et al., 2011) and have a direct influence on the lake's biological structure.

     Some non-conservative tracers such as fluorescent dissolved organic matter (FDOM), which can be determined using

parallel factor analysis (PARAFAC), have been used to trace dissolved organic matter (DOM) in aquatic environments (He et al., 2014; Massicotte and Frenette, 2011; Stedmon et al., 2003; Stedmon and Markager, 2005b; Walker et al., 2009). PARAFAC analysis is a modelling tool which can separate multiple FDOM samples (emission and excitation matrices) into specific fluorescent components (Stedmon et al., 2003). The fluorescent components can be biologically produced proteins derived from bacteria or molecules from the degradation of terrestrial organic material. These components have previously

been found visually using a single excitation emission matrix and the observed fluorescent peaks (Coble, 1996). The differentiation between the fluorescent components are both a strength and a weakness as we can isolate many different components, but all of them can differ in both degradation and production rate in the lake and groundwater. Furthermore, these FDOM components have not yet been investigated as tracers in groundwater-fed lakes, as they, just as the rest of the non-conservative biological tracers, are volatile.

This is observed as a change in tracer concentrations (often a decrease) after the groundwater is discharged to the lake. The speed of which the change in concentration occurs is typically related to seasonal variations (e.g. temperature, mixing of the water column and UV-radiation) and the WRT of the lake, e.g. the amount of time the tracer has been in the lake. The removal and degradation rates have been examined in many instances, e.g. for phosphorus (Larsen and Mercier, 1976; Vollenweider, 1975), nitrate (Harrison et al., 2009; Jensen et al., 1995), CDOM and DOC (Madsen-Østerbye et al., 2017). In

a modelling approach, these rates are important as they provide information about the change in tracer concentration from the time when the tracer entered the lake. From this, it is possible to back-calculate the mixed inflow concentration of specific tracers when they were discharged to the lake. These estimations are crucial when working with non-conservative tracers, as it enables a direct comparison between the tracer concentration found in the catchment and the estimated lake concentration before degradation took place, which originates from the mixed inflow of groundwater.

As the concentrations of both conservative and non-conservative tracers in a groundwater-fed lake correspond to the mixed concentrations of discharging groundwater, taking degradation and atmospheric deposition into account, it is possible to utilise the Community Assembly via Trait Selection approach (CATS). This model has been used to predict the relative abundances of a set of species from measures of community-aggregated trait values (average leaf area, root length etc.) for

all plant species at a site (Shipley, 2010; Shipley et al., 2006, 2011). The CATS model has three main parameters: (1) it models the probabilities that (2) maximize the entropy, which (3) is based on a set of constraints (Laliberté and Shipley, 2011; Shipley et al., 2011). In reality, the model (1) predicts the relative abundances of species at a location from their (3) average traits values by (2) minimizing the number of species that explain the mean traits values. Maximum entropy (2) is the maximizing of "new knowledge gained" related to plant communities. This means moving from "all species has the same relative abundances" to "a few species has a high relative abundance". When applying the model to the lake-groundwater interaction, we use the measured tracer concentrations at groundwater well sites around the lake as the individual plant species and the estimated mixed lake concentration before degradation took place as the community-aggregated trait values. Determining groundwater movement using both conservative and non-conservative tracers found around the lake shore overcomes some fundamental shortcomings related to traditional sampling. Firstly, we often measure tracers, which do not have a direct impact on the lake ecosystem and therefore do not provide meaningful information regarding the inflow of nutrients or CDOM. Furthermore, the sampling is only done in a few places throughout the catchment, which do not necessarily provide all the information on the groundwater flow patterns, or to which degree water enters the lake and where. To overcome this, we measured conservative and relevant non-conservative tracers in and around a small lake with the aim of developing a novel approach to identify groundwater discharge and recharge areas on a high spatial scale. Thus areas which deliver pollutants to the lake were pin-pointed, in which groundwater recharge happens and where recharge occurs with an increased flowrate. The latter can spark further investigations into the lake WRT. Information regarding the WRT of the lake is especially useful when investigating how the concentrations of pollutants in the lake will develop after future restoration attempts. In the present study, the eight following tracers was measured: FDOM, CDOM, DOC, total dissolved phosphorus (TDP), total dissolved nitrogen (TDN), total phosphorus (TP), total nitrogen (TN) and $\delta^{18}O/\delta^{16}O$ isotope ratios, and tested: (1) if groundwater discharge sites and pollutant sources can be estimated with the CATS model based on tracer concentrations, (2) whether conservative and non-conservative tracers can be used to detect groundwater recharge areas as well as provide insights into which areas possess a high groundwater recharge rate, and (3) if catchment-derived tracer concentrations can be used to estimate a range of WRTs, which can be used with the CATS model**.**

**Materials and methods**

A small groundwater-fed lake in the sandy northwestern part of Denmark was chosen for this study (Tvorup Hul, area: 4 ha, mean depth: 2.4 meters, 56º91 N, 8º46 E, UTM Zone 32). Coniferous forest and heathland dominate the catchment although some agricultural activities are found in the eastern part of the catchment (Fig. 1a). Various isoetids including the rare nationally threatened species *Isoetes echinospora* and *Subularia aquatica* inhabited the lake until some decades ago when brownification increased significantly (based on Rebsdorf, 1981 and the present study), probably due to increasing soil pH (Ekström et al., 2011). This led to a restoration attempt in 1992; a channel was established to bypass the stream going through the lake, thus making the lake groundwater-fed. CDOM, DOC and the hydrologic conditions in the lake have since

been investigated in several projects (Madsen-Østerbye et al., 2017; Solvang, 2016). This has resulted in extensive background data as well as estimations of WRTs between 0.4 and 3.3 years based on water table heights, hydraulic conductivity and seepage meter samplings (Solvang, 2016 and priliminary work P. Engesgaard personal communications, 2017).

**Sampling and laboratory analysis**

A total of 30 groundwater samples were taken every 50 meters around the lake within a distance of 5-45 m to the shore in temporary groundwater wells at 1.25 m depth in February 2016. The data preparation, analysis and results are visualised in figure 2. The water in the wells was replaced three times before transferring the sample water to an acid rinsed container. The samples were filtered through pre-combusted 0.7 μm nominal pore size Whatman GF/F filters the same day and kept cool and dark in hermetically sealed acid rinsed BOD flasks until examination. Unfiltered samples were also collected from the lake.

DOC concentrations were measured using a total organic carbon analyser (TOCV, SHIMADZU, Japan) in accordance with Kragh and Søndergaard (2004). The CDOM absorbance was measured on a spectrophotometer (UV-1800, SHIMADZU, Japan) between 240 and 750 nm in 1 nm intervals in a 1 cm quartz glass cuvette and expressed as the absorbance at 340 nm ($A_{CDOM}$(340) cm$^{-1}$). The samples were analysed for $\delta^{18}O$ and $\delta^{16}O$ isotopes at the Department of Geosciences and Natural Resource Management (University of Copenhagen) by mass spectrometry in accordance with Appelo and Postma (2005). $\delta^{18}O$ is presented in the standard $\delta$-notation V-SMOW as $\delta^{18}O$ ‰ (Vienna Standard Mean Ocean Water) (Turner et al., 1987). Total dissolved phosphorous (TDP) and total dissolved nitrogen (TDN) were determined for groundwater samples while total nitrogen (TN) and total phosphorous (TP) were determined for lake water, as inflowing nutrients become incorporated into aquatic organisms. Nutrients were measured by transferring 5 ml sample water and 5 ml potassium persulfate reagent to acid-rinsed autoclave vials before autoclaving for 45 minutes. Then 2.5 ml borate buffer was added after cooling and analysed in an auto-analyser (AA3HRAutoAnalyzer, SEAL, U.S.A) together with blanks and internal standard row.

**PARAFAC modelling**

The fluorescent properties of DOM samples were investigated using PARAFAC. The FDOM samples were initially diluted 2-12 times to account for self-quenching, also referred to as inner filter effect, which occurs with high CDOM absorbance in the sample (Kothawala et al., 2013). Sample and blank fluorescence were measured using a spectrofluorometer (Cary Eclipse, Agilent Technologies, U.S.A) by excitations between 240 and 450 nm, in 5 nm steps, while scanning the emissions from 300-600 nm in 2 nm increments. Prior to PARAFAC analysis, fluorescence data was processed in R (3.3.1) (R Core team, 2017) using the eemR (0.1.3) package. Blank values were subtracted following the documentation provided in the eemR package to remove Raman and Rayleigh scattering (Bahram et al., 2006; Lakowicz, 2006; Zepp et al., 2004). The data was then Raman normalised by dividing the florescent intensities by the integral of the Raman peak of the blank sample

(Lawaetz and Stedmon, 2009) and lastly corrected for the inner filter effect (Kothawala et al., 2013) before being exported to Matlab (2015b). In Matlab, the fluorescence data was combined with a larger dataset (>1000 fluorescent samples from Massicotte and Frenette (2011) originating from a range of diverse aquatic systems) in order to increase the diversity of FDOM components. This allows for the detection of components insufficiently represented in the collected samples (Fellman et al., 2009; Stedmon and Bro, 2008; Stedmon and Markager, 2005a). The drEEM package was used to do the PARAFAC modelling following the same procedure as described in Murphy et al. (2013). A split–half analysis, in which the dataset is split into two parts and compared multiple times, were used to test the results found in the PARAFAC model. A contour map showing the measured FDOM concentrations in groundwater was plotted in ArcMap (ArcMap 10.4.1, ESRI, U.S.A) using the inverse distance weighted (IDW) function with barriers fitted around the lake and drainage channels.

**Groundwater recharge and areas with a high groundwater recharge rate**

A hierarchical Euclidean cluster of $\delta^{18}O$ ‰ was employed to determine groundwater recharge areas using the Stat base package in R. $\delta^{18}O$ ‰ was chosen as it is both conservative and biologically inert. Groundwater well sites which formed a cluster together with the lake samples were considered as being groundwater recharge sites, e.g. water originating from the lake, and was excluded for the later estimations of groundwater discharge sites. The groundwater recharge sites were further investigated using a range of non-conservative tracers influenced by biological degradation. We found that some of the tracer concentrations changed when moving from the lake to the groundwater. For example, CDOM showed a decrease in concentration when entering the groundwater, which is properly due to pH changes in the soil. An inspection of the results revealed that a protein based fluorescent component met our criteria of being (1) non-conservative, (2) not afflicted by the lake groundwater interface and (3) not too easily degraded or produced in high amounts, which could create false positive groundwater recharge sites. The PARAFAC component was related to the lake concentration with a Hierarchical Euclidean cluster dendrogram and the sites which clustered together with the lake samples indicated a high groundwater recharge rate.

**Non-conservative tracer degradation and lake WRT**

Lake WRT was found using traditional hydrological methods combined with non-conservative tracer concentrations, which were related to their degradation rates to form a proxy for the maximum WRT. Previous hydrological models suggested that the lake had a WRT between 0.4 and 3.3 years. To further narrow this range, we estimated WRT by relating the concentrations found in the lake to their respective degradation rates related to increasing WRT, e.g. by adding the estimated removed tracer since the groundwater entered the lake to the measured concentration in the lake. This enabled us to narrow the span of the WRT based on the estimated mixed inflowing tracer concentration related to the actual catchment concentrations. E.g. if the estimated inflow concentration of a tracer is 100 µmol l$^{-1}$ at a WRT of 2 years, and the highest catchment tracer concentrations is 50 µmol l$^{-1}$, then the catchment does not support a WRT of 2 years. In this instance, we estimated lake tracer concentrations of TN, TP, CDOM and DOC for WRTs from 0.25 to 3.5 years in 0.25 year increments following Eq. (1):

$$MIC = \frac{tr_{lake}}{ret\,(frac)}, \tag{1}$$


where *MIC* is the mixed inflow concentration, $tr_{lake}$ is the tracer concentration found in the lake and *ret (frac)* is the retention fraction of the tracer at a known WRT. Retention models used in this study were based on the lake type as well as the geographical location of our lake. As there is not one model that can provide removal rates across all lakes, we encourage the readers to find models related to their specific lake type. Thus, phosphorus equilibrium concentration in this study were

found using Eq. (2) modified from Larsen and Mercier (1976) which describes phosphor retention in lakes with low productivity:

$$retP\,(frac) = 1 - \frac{1}{1+\sqrt{WRT}}, \tag{2}$$

where *retP* (frac) is the retention fraction of phosphorus and *WRT* is the water retention time in the lake. Similarly, nitrate inflow concentration were estimated using a modified Danish nitrate removal model derived from Jensen et al. (1995)

describing retention for shallow lakes with a short WRT (0-6 years) Eq. (3):

$$retN\,(frac) = \frac{59 \cdot WRT^{0.29}}{100}, \tag{3}$$

where *retN (frac)* is the retention fraction of nitrate and *WRT* is the water retention time in the lake. The corresponding retention fractions removed at different WRT were related to the lake concentrations to estimate what the mixed inflow concentration must have been to produce the present lake concentration. The combined summer UV-radiation and bacterial

degradation rates of DOC and CDOM in groundwater from the dominating catchment vegetation type of the lake (Madsen-Østerbye et al., 2017) were extrapolated to the rest of the year. This was done by relating the degradation rates to the mean monthly UV-index (DMI, 2015) while assuming a linear relationship between the UV-index and degradation rates. Thus we were able to estimate the specific removal of DOC and CDOM on a monthly basis related to the concentration measured in the lake at the time of sampling following Eq. (4):

$$tr_{lake} = tr_{lakepm} - tr_{lakepm} \cdot degra\,(frac) - tr_{lakepm} \cdot mf + tr_{inflow} \cdot mff, \tag{4}$$

where $tr_{lake}$ is the lake concentration in the specific month, $tr_{lakepm}$ is the lake tracer concentration in the previous month, *mff* is the monthly flushing fraction (mff = 1/WRT/12), *degra (frac)* is the degradation fraction in present month related to UV-radiation and $tr_{inflow}$ is the inflowing tracer concentration. Eq. 4 was solved for $tr_{inflow}$ and calculated using the same WRTs as the nitrate and phosphorus models.

**The CATS model**

Since the concentrations of both conservative and non-conservative tracers in a groundwater-fed lake correspond to the mixed concentrations of discharging groundwater, while taking degradation and atmospheric deposition into account, it is possible to utilise the CATS model. In the present study, the concentrations of non-conservative tracers (DOC, CDOM, TDP and TDN) at groundwater well sites around the lake acted as the individual plant species at a site and the equilibrium tracer

concentrations derived from Eq. 1 (DOC, CDOM, TP and TN) acted as the community-aggregated trait values. When

choosing tracers, it is important that there is differentiation between the concentrations measured at the sites. This means that a higher number of tracers and higher un-correlated concentration differences between the sites results in a more secure determination of groundwater discharge sites. All tracers were investigated as a combined package, e.g. a single site is described by all the tracers mentioned above, and was run using the maxent function in the FD (functional diversity) package

in R to compute the CATS model (Laliberté and Shipley, 2011). Further information on the calculations can be found in the supplementary material for the FD package (Laliberté and Shipley, 2011). From this, the model predicts the minimum number of groundwater well sites along the lake shore, which explains the measured concentrations in the recipient lake by maximising the sites' relative contribution. The model also computes lambda values from the least squares regression measuring which tracers are most influential on the relative fractions of water originating from the groundwater well sites.

Consequently, Lambda values quantifies how much the relative contribution from the sites change when one tracer is changed a unit while the rest of the tracers are kept constant.

## Results

### Groundwater recharge

Recharge areas were identified by a Euclidean hierarchical cluster dendrogram of $\delta^{18}O$ ‰. The cluster revealed two main

groups marked with orange and light-blue in figure 3. The first group (orange) shows the groundwater well sites, ranging from site 18 to 29, which clustered together with lake samples. The samples in this orange cluster share a clear resemblance with lake $\delta^{18}O$ ‰ measurements and were therefore considered as groundwater recharge sites. The recharge sites were located in the north and western part of the lake and are marked with orange in figure 1a.

### Fluorescent DOM

PARAFAC and split-half analysis modelling identified five distinct fluorescent DOM components (C1-C5, explained variance 96.79 %). The spectral properties of the five fluorophores (components) identified by the PARAFAC analysis (Fig. 4) revealed that the DOM pool had both terrestrial and microbial influence. Component C1 was similar to previously found components from terrestrial humic-like material (Stedmon et al., 2003). The component absorbs in the UV-C region, which is absorbed by the ozone layer and atmosphere (Diffey, 2002), and is for this reason expected to be mainly photo-resistant

(Ishii and Boyer, 2012). Component C2 has been reported to be both marine and terrestrial humic-like (Coble, 1996; Murphy et al., 2006) and seems to be degraded by visible light and produced by microbial degradation in equal amounts (Stedmon and Markager, 2005b). Component C3 was also believed to be of terrestrial humic-like origin and was similar to the fluorescent peak C described by Coble (1996). The component absorbs in the UV-A region and is susceptible to both microbial and photochemical degradation (Stedmon et al., 2007; Stedmon and Markager, 2005a). Component C3 may be an

intermediate product or produced biologically since changes in the concentration have been observed in the open oceans and in sea ice that has no apparent connection to the terrestrial environment (Ishii and Boyer, 2012). Component C4 is similar to

component 5 found in Stedmon et al. (2003) and is believed to be a combination of fluorescent labile materials named peak N and T which are produced biologically associated with DOM degradation (Coble, 1996; Stedmon and Markager, 2005b). Component C5 is linked to free tryptophan, which is a product of microbial activity (Determann et al., 1998). This component has been found to decrease during dark incubations and UV exposure (Stedmon et al., 2007), but component C5 is also associated with the degradation of DOM (Stedmon and Markager, 2005b) and autochthonous production (Murphy et al., 2008).

The highest fluorescence concentrations were found in the groundwater while the lake water fluorescence concentrations were generally lower (Table S1). Component C1 had the highest fluorescence with a value of 7.8 Raman's units (R.U.) in the lake and a maximum fluorescence of 47.1 R.U. at groundwater well site 7. Component C5 had the lowest fluorescence in the lake (0.27 R.U.) and a maximum fluorescence of 2.9 R.U. at groundwater well site number 8. Component C5 also varied much between groundwater samples with the lowest value of 0.1 R.U. or 28 times lower than the maximum concentration. Components C1, C2 and C3 had low lake-like concentrations in recharge areas (orange sites in figure 1a). Concentrations of C4 were generally higher in groundwater around the lake than in the lake (1.1-1.5 versus 1.1 R.U. visualised in figure 1b). Component C4 was chosen as a proxy for groundwater recharge as the concentration of the C4 component increase with biological activity and there was no apparent concentration changes in the lake-groundwater interface. The cluster diagram of component C4 showed that especially site 24 grouped with lake samples, but sites 20, 21, 23 and 26 also showed high comparability with the lake (Fig. 5), which can also be observed from the IDW map of component C4 around the lake (Fig. 1b).

**Groundwater discharge areas and lake WRT**

Tracer concentrations of the lake narrowed down the possible WRT. Equilibrium tracer concentrations of DOC, CDOM, TDP and TDN (found using Eq. 1-4) for water retention times between 0.25 and 3.5 years in 0.25 increments revealed that concentrations of TDN in the catchment are not high enough to support WRT values over 2 years based on the nitrogen retention model used. Thus, catchment tracer data revealed a possible WRT between 0 and 2 years.

Groundwater discharge areas were found employing the CATS model on nutrient concentrations and dissolved organic matter fractions estimated using Eq. 1. related to WRTs between 0.25 and 2 years. The estimated phosphorus concentrations ranged from 46 to 80 $\mu$g P l$^{-1}$ (Eq. 1 and Eq. 2) while nitrate concentrations ranged from 1113 to 2417 $\mu$g N l$^{-1}$ (Eq. 1 and Eq. 3). CDOM and DOC degradation rates were related to the UV-index and varied from 0.64 % in December to 28 % per month in June for DOC and between 0.46 % and 20 % for $A_{CDOM}(340)$ in the same months and were significantly different from each other (P < 0.001). Thus, estimated mixed inflow concentrations of CDOM ranged from $A_{CDOM}(340) = 0.43$ to 1.04 cm$^{-1}$ while DOC ranged from 1205 $\mu$mol l$^{-1}$ to 3160 $\mu$mol l$^{-1}$ for WRT between 0.25 and 2 years (Eq. 1 and Eq. 4). The CATS model isolated the minimum number of sites that explained the estimated lake concentrations. The model identified the following sites: 1, 9, 11, 13 and 14 as the groundwater discharge sites delivering more than 0.1 % of the water throughout the different WRTs (Fig. 6). Changes in site distribution and fractions of discharging water were observed between the

different WRTs, but in general, groundwater from 3-4 sites explains the estimated concentrations in the lake (Fig. 6). Site number 14 delivers more water with higher WRT (to a maximum of 54 % of the total discharge), site 1 peaks at a WRT of 1.25 providing 27 % of the water to the lake, site number 9 delivers less water with increasing WRT, but 49 % at the lowest WRT of 0.25 years. Site number 11 delivers 26 to 34 % of the water to the lake until a WRT above 1.5 years is reached, where site 13 explains the concentration in the lake better and provide 29 and 19 % of the water to the lake. Overall 73 to 96 % of the water is estimated to arrive from the eastern part of the lake, while site number 1 (in the southern part) is estimated to deliver 4 to 27 % of the water. Lambda values, explaining which tracers are the most important when predicting the fractions of water origination from groundwater well sites, showed that CDOM was the most important tracer when determining which sites delivered water to the lake with a mean lambda value for all WRTs of 24.2 versus 0.1-5.9 for the other tracers.

**Discussion**

The combination of biological and hydrological methods in a novel approach provided a better estimate of the WRT, an identification of groundwater recharge and discharge areas as well as the fractions of water coming from each site. Based on the model results and earlier hydrological studies, we will discuss the main questions from the introduction (1) if groundwater discharge sites and pollutant sources can be estimated with the CATS model based on tracer concentrations, (2) whether conservative and non-conservative tracers can be used to detect groundwater recharge areas as well as provide insights into which areas have a high groundwater recharge rate and (3) if catchment-derived tracer concentrations can be used to estimate a range of the water retention times, which can be used with the CATS model. Furthermore, we will discuss which of the tracers work and which could possibly work with refined methods. And how these findings could benefit lake restoration programmes.

**Determination of groundwater recharge areas**

Groundwater recharge sites were identified along the northern and western part of Tvorup Hul with a hierarchical cluster analysis of the conservative $\delta^{18}O$ tracer. The exact same areas are also the ones with adjacent drainage channels (Fig. 1a), which facilitate the areas as recharge sites. While $\delta^{18}O$ ‰ worked well as a general groundwater recharge estimator, it does not indicate which sites deliver more water. An indication of this can be found when examining the non-conservative tracers such as the fluorescent components.

Sites resembling the fluorescence found in the lake will indicate a high groundwater recharge rate, while a difference in concentration between lake and groundwater sites will indicate a lower groundwater recharge rate where there is sufficient time for a significant modification of the components representing the DOM pool of the lake. The fluorescent component C4 has previously been found to increase with biological activity (Coble, 1996), which is why we used it as a proxy to estimate the sites with high groundwater recharge rate. The hierarchical Euclidean cluster dendrogram of component C4 showed that

sites in the northern part of the lake formed a group (sites 24, 20, 21, 23 and 26) (Fig. 5 and visually in Fig. 1b). This information can be of importance related to placement of seepage meters, which will result in better estimations of the groundwater discharge and recharge and as such the modelled WRT of the lake. In other words, it might be advantageous to carry out groundwater sampling first to estimate sites with high discharge rates, then estimate WRT utilising these sites and finally model groundwater discharge areas by using the improved and narrowed WRT range.

CDOM generally showed much lower absorbance at groundwater recharge sites than in the lake making it less suitable for estimating recharge areas. The decrease in absorption is possibly due to low soil pH causing flocculation of CDOM in the soil matrix (Ekström et al., 2011). The same was observed with fluorescence of component C1, which had lower intensities in recharge areas, indicating that component C1 is linked to CDOM. While component C1 was not particularly useful for estimating groundwater recharge, it could be useful to estimate discharge sites. To utilise the component for discharge estimates there is a need for an assessment of the degradation rate. While it has been shown that component C1 is largely photo-resistant, as it does not absorb the UV-A radiation areas, and is largely resistant to microbial degradation processes (Ishii and Boyer, 2012), no reliable rates for the degradation have been found. In this study, we found that only sites number 9 and number 11 hold concentrations lower than the lake (Table S1) indicating that most groundwater discharge would originate from these sites if little to no degradation takes place.

**Determination of groundwater discharge areas**

Neither $\delta^{18}O$ nor previous seepage meter samplings have achieved a similar understanding of groundwater recharge areas in Tvorup Hul as compared to the present approach. While the $\delta^{18}O$ ‰ provides a way of separating groundwater and surface water, using it to determine groundwater discharge sites is simply not possible due to the homological distribution seen in groundwater (Krabbenhoft et al., 1990). Previous seepage meter samplings provided scattered and momentary estimations of discharge sites, indicating that groundwater entered the lake from the southern bank (Solvang, 2016). This does not correspond to tracer concentrations found in the southern area, which show very high CDOM absorbance at 340 nm ($A_{CDOM}(340)$ = 1.3-3.1 cm$^{-1}$) and DOC concentrations (3114-10467 µmol l$^{-1}$) in relation to the lake ($A_{CDOM}(340)$ = 0.4 cm$^{-1}$/DOC 1058 µmol l$^{-1}$). This hints that the lake is influenced by groundwater discharge from other areas as well. The lowest DOC concentrations in the southern area were several times higher than those from the equilibrium estimation suggesting a WRT above 6 years which is well beyond previous estimates of WRT. Samples from the eastern shore had lower concentrations all around proposing that water from this area influence the lake water. Thus, if the water actually originated from the southern area, the lake would need to have a prolonged WRT resulting in increased removal of tracers from the lake. This requirement conflicts with the remaining tracers, where especially TDN sets an upper limit to the WRT of 2 years. The CATS model used in this study shows that while a fraction of groundwater enters the lake from the southern bank, most of the water originates from the eastern shore (Fig. 1a). Seepage meter measurements from the eastern shore showed both discharging and recharging of groundwater (Solvang, 2016). The same was observed for $\delta^{18}O$ ‰ samples from the eastern part of the lake, which were lower than in the groundwater from the southern shore, indicating an influence of newly

precipitated water or influence from the lake. Sampling in the northeastern and eastern part of the lake revealed an area with little groundwater and a clay deposit layer which possibly reduces infiltration to deeper groundwater layers. As a result, precipitations could enter the lake as surface and subsurface runoff water originating from the hills to the east and the plateau in the northeastern corner (Fig. 1a), resulting in short bursts of discharging water. The multi-tracer approach enables the determination of discharge areas much more precisely and on a temporal scale related to the WRT of the lake (in this instance the previous 3 to 24 months). Consequently, the model is able to track uncommon events such as heavy precipitation events in which large amount of water is discharged to the lake during a short period. These events are often difficult to track as seepage metres need to be deployed in this exact period as well as in the right places.

**Tracer influences**

Most tracers used in this study are less conservative compared to $\delta^{18}O$ and can therefore change both in the lake water and in the catchment soils. This entails a minimum understanding of processes and rates that influence the concentrations. The temporal variability in nitrate concentrations in groundwater are related to the flow rate rather than seasonal changes (Kennedy et al., 2009). The same was observed for phosphorus, where particularly dry periods followed by heavy rain increased the phosphorus concentration measured in groundwater-fed springs (Kilroy and Coxon, 2005). Thus, in the case of northern Europe, sampling during late winter might be the best solution because soils are saturated at this time of year (Sand-Jensen and Lindegaard, 2004). Previously polluted areas, e.g. from wastewater infiltration, with increased concentrations of DOC and nutrients are likely to be in a state of imbalance resulting in a reduction in concentrations over time (Repert et al., 2006). For this reason, in these areas, it is important to have temporal sampling following decreases in concentrations and to relate them to lake concentrations during sampling. Lake inter-annual DOC and CDOM changes were generally low in our study with an annual $A_{CDOM}(340) = 0.41$ cm$^{-1}$ ± SD 0.05, corresponding to what is observed in larger water bodies where WRT integrates inflowing DOC and CDOM (Winterdahl et al., 2014). Inter-annual DOC and CDOM variations in groundwater from the lake catchment (Fig. S1) showed the same tendency as described for nutrients, and this suggests that sampling should be done at multiple times or in a period without drought or high rainfall. On a broader scale, the variation in DOC is known to be related to hydrology (Erlandsson et al., 2008), mean air temperature (Winterdahl et al., 2014) and the recovery from acid deposition (Evans et al., 2006; Monteith et al., 2007). Sampling from wet areas with standing surface water resulted in high concentrations of most tracers (Table S1). Consequently, these areas should be avoided seeing that they provide no information regarding the discharge of groundwater. The removal of CDOM and DOC also changes on an annual basis in lakes and is related to bacterial degradation, photo-degradation, sources and mixing of the water column. A sensitivity analysis of the results was conducted by running the CATS model with a ± 10 % change in tracer concentrations. The results showed that sites generally remained unchanged with only smaller deviations in percent wise distribution in discharge up to a WRT of 1.25 years (Fig. S2). Above this point, there are some differences in sites, which change between sites number 11 and number 13. In conclusion, even when changing multiple parameters in the model, the same five

groundwater wells are identified explaining the measured lake concentrations. Future investigations into variation of tracers in groundwater and degradation rates in lakes will likely strengthen this model.

The processes that influences changes in FDOM are still being investigated (Ishii and Boyer, 2012). Tracing FDOM has been done in both rivers and open waters (Baker, 2001, 2002; Stedmon and Markager, 2005a), but only a few studies have
been conducted in groundwater. These studies have focused on changes in FDOM from deep groundwater wells (Lapworth et al., 2008) or tracing FDOM using samples that are collected very far apart (Chen et al., 2010). Specific fluorescence intensity of components showed large differences among sites in this study, up to a factor of 28, between groundwater well sites, lowest at site 11 and highest at site 8, around the relatively small lake. These findings illustrate the problem when applying FDOM as a tracer over large distances in groundwater. Besides bio- and photo-degradation of fluorescent
components, absorption changes have also been observed in relation to Fe(III) concentrations (Klapper et al., 2002). This might change the concentrations of FDOM components as they travel from anoxic groundwater with reduced iron into the oxic lake water. Overall, PARAFAC components have the potential to work as groundwater tracers, but there is a need for a better understanding of the processes that cause changes in fluorescence characteristics of DOM and hence concentrations of FDOM components both in the lake and in the lake-groundwater interface.

**Potential lake management influence**

The determination of discharge sites can result in direct management related to specific problematic areas. The model used in this study showed that water entering the lake primarily originated from the catchment to the east of the lake. If water from this part was diverted around the lake, there would be a reduction in CDOM absorbance of 61-89 % based on calculations relating percent-wise discharge, its concentrations and WRTs from 0.25 to 2 years in 0.25 increments. On the
contrary, diverting water around the lake at site number 1 would only result in a lowered inflowing CDOM absorbance of 11-39 %. Moreover, in both cases, there would be an increase in photobleaching of present CDOM in the lake caused by the increased WRT. Furthermore, huge reductions would occur for TP and TN, with a decrease of 82-96 % if diverting water from the eastern shore, in contrast to the southern shore with a modelled decrease of 4-18 % in TP and TN. In the future, hydrology is likely to be the main driver of variability in DOM (Erlandsson et al., 2008) with an estimated increase in
CDOM by a factor 4 in lakes with short WRT (Weyhenmeyer et al., 2016). This makes it critical to establish a modelling tool that is capable to pinpoint sites delivering pollutants to lakes, and provide us with the ability to take action and reduce the impact on the ecological state of lakes.

**Conclusion**

The present method and modelling tool can improve estimates of recharge and discharge areas as well as WRT in smaller
lakes on a temporal scale. The model provides accurate estimates of discharge fractions, related to field measurements, and can be used for precise management of problematic pollution areas. The hierarchical clustering can be used to estimate groundwater recharge sites, which can be incorporated as a guideline for a better estimation of WRT in lakes. Furthermore,

the use of multiple tracers strengthens the model and keeps a certain degree of freedom in regard to the choice of tracers related to laboratory capabilities.

**Acknowledgements**

We are grateful to Naturstyrelsen (The Department for Nature and Conservation) for access to the study area in Nationalpark Thy. The study was partly supported by a grant to the Centre for Lake Restoration, a Villum Kann Rasmussen Centre of Excellence. Peter Engesgaard and Ingeborg Solvang are acknowledged for their comments on the manuscript and for supplying data on oxygen isotopes.

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

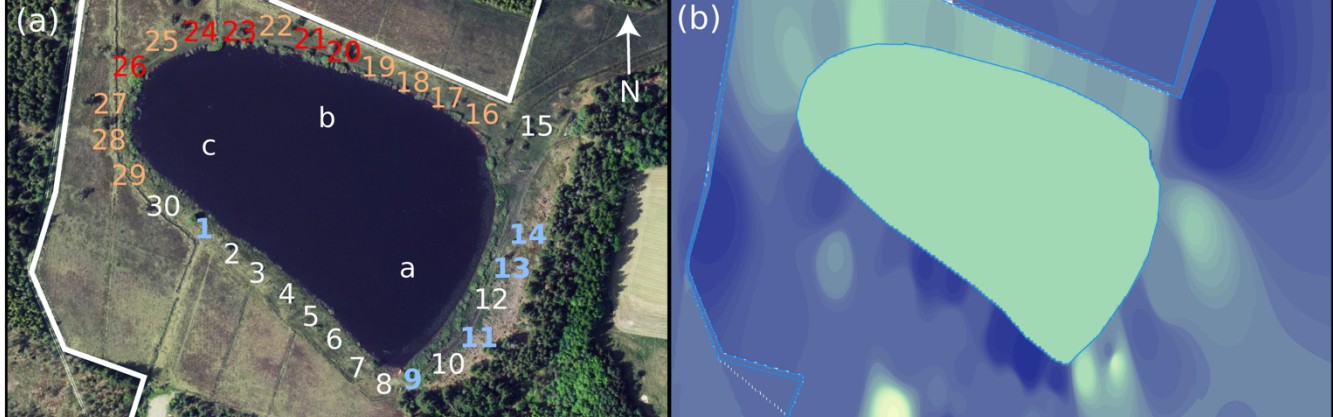

**Figure 1: (a) Aerial photo of Lake Tvorup Hul showing groundwater well sampling sites (numbers). Orange numbers denote groundwater recharge sites, red numbers show sites with a high degree of recharge, white numbers represent possible groundwater discharge sites and light blue shows model isolated discharge sites. Positions a, b and c show the three sampling sites in the lake. Missing samples in the northeastern part are due to an absence of groundwater in the area. The adjutant drainage channels north and west of the lake are marked with white lines. (b) Inverse distance weighted (IDW) contour map of fluorescence**
**component C4. Bluegreen colour corresponds to lake concentrations; darker blue indicates increased concentrations and lighter blue denotes decreased concentrations throughout the catchment. Areas with low differences between fluorescence in the lake and in the catchment are seen north of the lake and indicate parts with a fast groundwater recharge.**

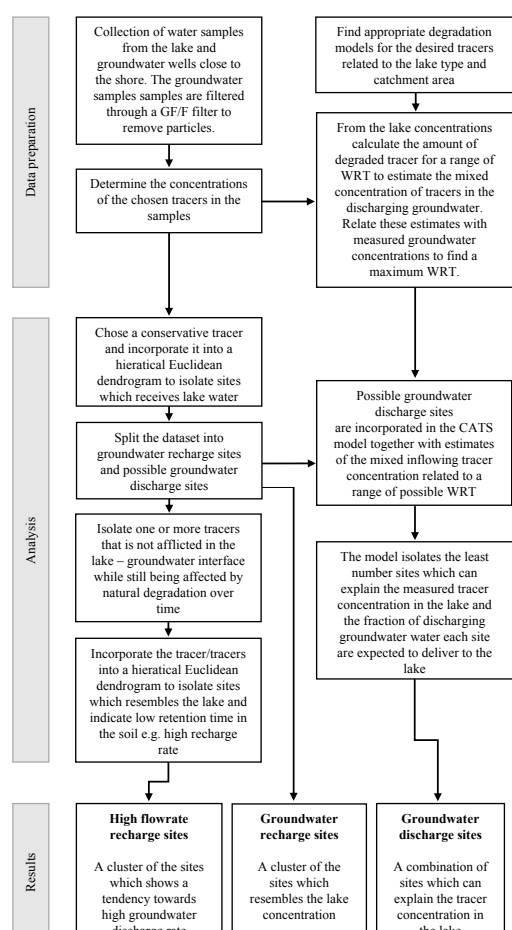

**Figure 2: Diagram showing the workflow from data preparation, analysis to the results.**

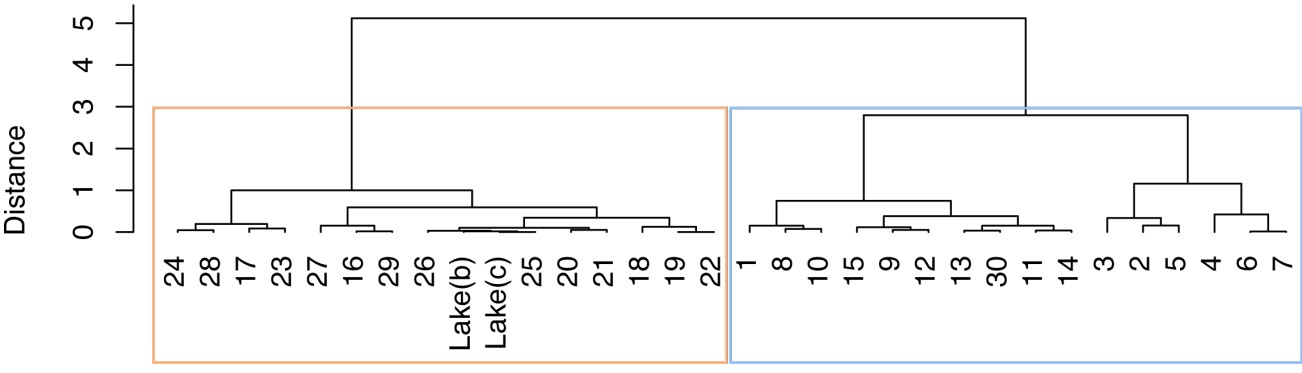


**Figure 3: Euclidean hierarchical clustering of the $\delta18O$ ‰ showing two clusters. First cluster, marked with orange, groups with lake samples and are therefore regarded as recharge sites. The other cluster, marked with light blue, is possible groundwater discharge sites to the lake. The y-axis denotes the linear distance between the $\delta18O$ ‰ samples fed to the model. The third lake sample was lost during preparation.**

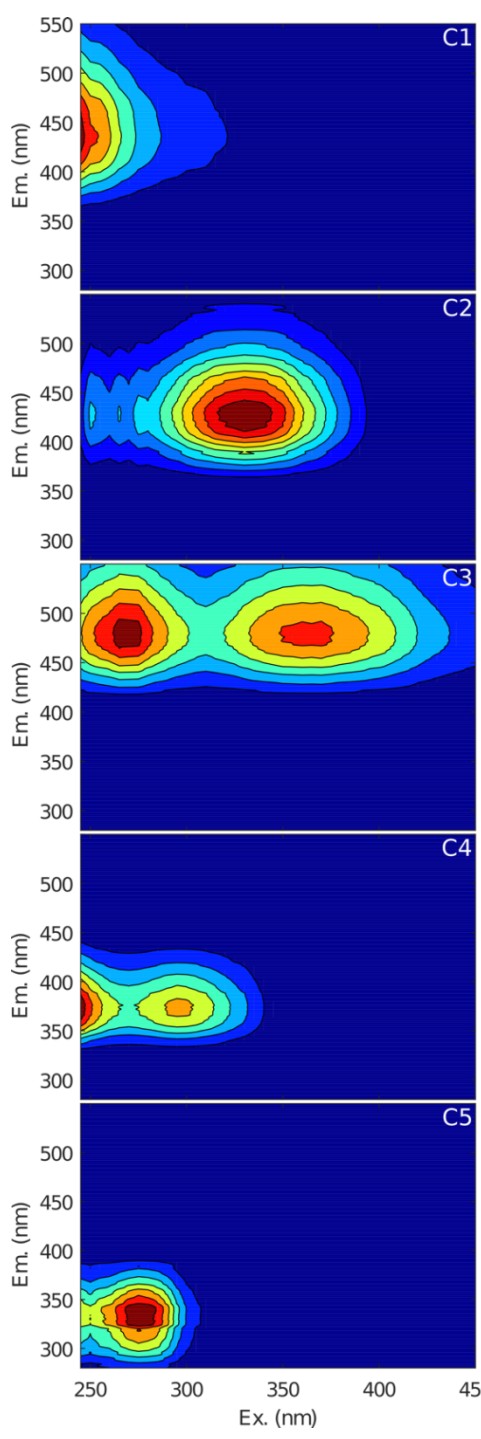


**Figure 4: Spectral properties of the five PARAFAC components (C1-C5) found in this study. The x-axis shows the excitation (Ex) wavelength in nanometre (nm) and the y-axis shows the emission (Em) wavelength in nanometre (nm) with low fluorescence being blue and high being red.**

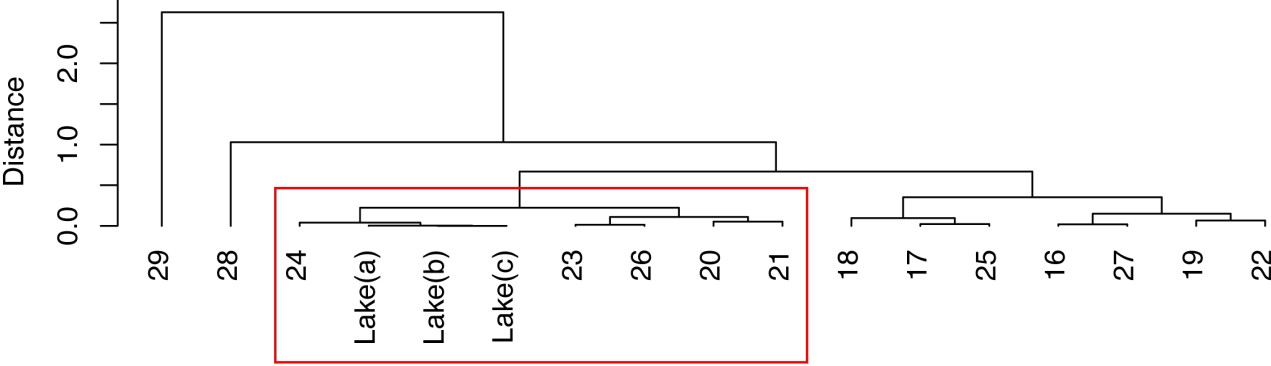

**Figure 5: Euclidean hierarchical clustering of fluorescent component C4 from recharging groundwater sites. The fluorescence found at sites 20, 21, 23, 24, and 26 clusters together with lake fluorescence (marked red). This indicates that these sites have a high degree of groundwater recharge. Groundwater well site 24 seems to be especially important in this regard.**

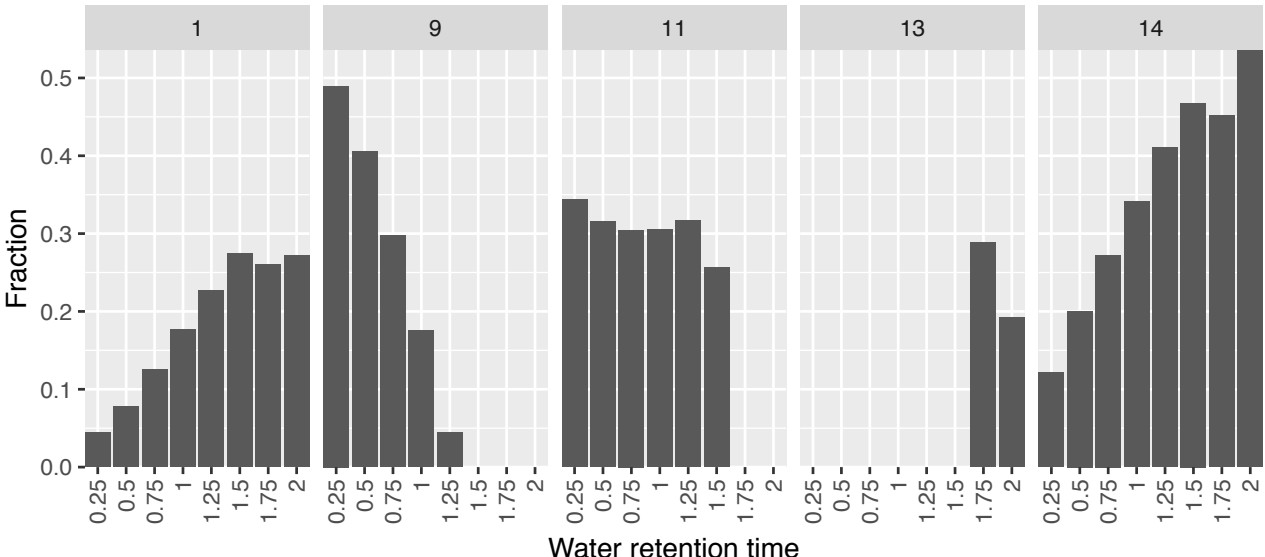

**Figure 6: Results derived from the CATS model shown in a bar plot in which the groundwater well sites (their numbering) are seen on the top x-axis and the fractions of groundwater discharge estimated to derive from the sites at the y-axis (only the sites which are delivering more than 0.1 % water to the lake are shown). The bottom x-axis denotes the different water retention times used in this model. Three to four sites generally explain the estimated concentrations in the lake.**