# Peer review of "Catchment tracers reveal discharge, recharge and sources of groundwater-borne pollutants in a novel lake modelling approach"

_Biogeosciences, 2017_

## Referee Comment (RC1) · Anonymous Referee #2 · 31 Oct 2017

**General Comments**

1. The authors need to spend more time on proofreading the manuscript. In its current state, the grammar does not hold up to the standards of Biogeosciences unfortunately. I have included a list of 27 items in technical corrections *for only the first five pages* of the manuscript. I encourage the authors to find a colleague to proofread or to use a reviewing service before the next submission.

2. In the paper, you are trying to combine several methods in a novel way to characterize the lake hydrology. Because of this, the methods must be clearer and not assume that the readers are familiar with the various components. The purpose of a certain method, the details of the models used, the interpretation of results needs to be more extensive. For example:

    a. The PARAFAC analysis: How do you get/What is the interpretation of the results in Figure 3?

    b. The determination of WRT is unclear still. The text (Line 156) briefly describes that WRT was estimated through tracer concentrations if no degradation took place. There is no further mention beyond this paragraph and needs to be elaborated upon.

3. I have some reservations on the approach taken with determining the groundwater discharge areas and lake WRT (section starting from line 215) or the methods have not been described adequately. I appreciate that this section was added but it is still not clear. When reading the methods, the reader needs to be able to see how you took your data and processed it, and arrive at the resulting figure or results. Additionally, I am not convinced that the equations used in this section were used correctly (see specific comments below).

**Specific Comments**

Line 111 - The PARAFAC analysis was described as a three-way modelling tool but it is not clear between what three things

Line 149 – What are the lambda values here? You should briefly comment on what this is

Line 153 – The text is describing the degradation of tracer concentrations in the previous sentence. Then it is followed by "This equilibrium estimations"… Is this referring to equilibrium concentrations? Degradation rates?

Line 165 – The Vollenweider equation that you provide as equation 2 is not the form provided in the 1975 paper you cite and has been used erroneously. From a mass balance approach, the mass fraction that is exported ($C_{outlfow}/C_{initial}$) = $1/(1+k*WRT)$

- If you have decided to replace the removal rate constant k with $WRT^{0.5}$, you need to justify this with further literature

- The general form of the equation that is presented in your paper looks to be the percent *export*, rather than percent *retention.* You will have to use $1-(1/(1+k*WRT))$ to get retention

- In fact, the form given in this manuscript shows that the percent retained increases with decreasing WRT.

Line 169 – I find it difficult to see how equation 3 is applicable. The equation only applies for lakes that have a WRT of approximately 0 – 6.2 years (and I agree the study site fits

inside this range). However, the source that was cite for this equation is in a Danish report published over twenty years ago and is not easily accessible. As an empirical equation, it is extremely difficult for the reader to understand the limitations, assumptions and the overall validity of this equation for this study (e.g. anyone trying to replicate your methods)

Line 178 – It is unclear if the authors developed equation 4 on their own or was from literature and thus should be clarified. Regardless, it appears that this equation is not internally consistent with respect to units. The monthly flushing rates are in units of 1/Time; thus the first two terms on the right hand side of the equation are in concentration units, whereas the last two terms are in concentration/time

Line 180 – Is the peak degradation through UV-radiation determined linearly from the UV-time relationship? I.e. is the peak degradation of 100% assumed to coincide with the peak UV radiation?

Line 200 – "…combination of peaks N and T produced biological (Coble, 1996)." does not make sense. What Ns and Ts are you talking about? What biological is produced?

Line 218 – The conclusion that the concentrations of TDN do not support a WRT value of over 2 years is wholly dependent on the model you choose. With the limited support provided for the model, this claim is not strong.

Line 297 – I do not think the manuscript does an adequate job in convincing the reader that this method can capture uncommon or stochastic events. It did not seem like the samples taken were taken at different times of the year when extreme conditions occur. Is there literature that supports your claim that these environmental tracers capture these stochastic events? Even so, how would a single sampling campaign be used to extrapolate beyond the timeframe or snapshot of when you sampled?

Line 320 – I'm not sure that claiming the analysis remaining generally unchanged by running the CATS model with a 10% perturbation of tracer concentrations is sufficient. The tracer concentrations in your supplementary material show that TP, TN and DOC all can vary by an entire order of magnitude between sites. Can you be certain that they also cannot fluctuate by an order of magnitude throughout the year?

**Technical Corrections**
1. Line 17 – WRT has not been introduced yet, do not use abbreviation
2. Line 17 – WRT was estimated to _be_ 2 years
3. Line 17 – Isolation of groundwater recharge areas _was_
4. Line 18 - …sites with _a_ high degree of recharge _were_
5. Line 29 - …which to some _degree_
6. Line 30 - …the groundwater contributes nutrients
7. Line 37 - …_particularly_ in small water bodies. _For example_
8. Line 53 – $^{36}Cl$
9. Line 60 - You are talking about the nutrients (plural) so it should be "which _are_ either remineralized when dying…". Also poorly phrased as nutrients (which is the subject of this sentence do not die)
10. Line 66 – _fluorescent_

11. Line 75 – use consistent style for lists throughout your paper. Either use (1) as you did in line 52 or keep to 1). Also use a colon to introduce your lists, not a semicolon
12. Line 84 – Subularia _aquatic_
13. Line 90 – _preliminary_ work
14. Line 93 – within 5-45m of what?
15. Line 96 – _hermetically sealed_
16. Line 100 – _quartz_
17. Line 101 – $\delta^{18}O$ is a ratio, not a concentration, please fix this throughout your paper
18. Line 107 – borate buffer _was_
19. Line 117 – were subtracted _… to remove_
20. Line 117 is a run on sentence; separate is at "the data were then Raman normalized…"
21. Line 129 – _biologically inert_
22. Line 134 is not a full sentence
23. Line 137 – atmospheric _deposition_
24. Line 139 – …linear in features. What features are you talking about?
25. Line 142 – What is the FD package?
26. Line 143 – In _the_ present study
27. Line 147 – The model outputs maximum entropy _probability_ fractions

---

## Referee Comment (RC2) · Anonymous Referee #1 · 6 Nov 2017

The authors applied different potential tracers in order to improve the reliable identification of recharge and discharge areas around lakes. Their main point is that they apply a model which is they claim to enable a quantitative determination of groundwater discharge of single sites based on tracer concentrations. Similar to the other reviewer I have several concerns about the manuscript's structure and concept. Overall I have to say that it is quite hard to judge the scientific quality of the work by now since too much is unclear to me and I am unable to evaluate if this is due to poor writing/style or due to an actual lack of a scientifically profound basis of the study. However, it is quite clear that a lot of effort was put into the investigations, especially regarding analysis of samples and data which at least justifies granting the authors the option

to revise their manuscript. Having said this, I strongly recommend being much more specific in the description of the approaches and methods applied. In many parts, the manuscript is extremely vague and statements and descriptions are kept far too general. My major concern about the study is the fact that mass concentrations and hydrology do not necessarily correspond linearly as the model assumes. This means that it is not possible to relate high tracer concentrations to high groundwater discharge rates. The authors have not made clear how the model overcomes this problem.

Please also note the supplement to this comment:
https://www.biogeosciences-discuss.net/bg-2017-209/bg-2017-209-RC2-supplement.pdf

---

## Author Comment (AC1) · 28 Dec 2017

**General Comments**

1. The authors need to spend more time on proofreading the manuscript. In its current state, the grammar does not hold up to the standards of Biogeosciences unfortunately. I have included a list of 27 items in technical corrections *for only the first five pages* of the manuscript. I encourage the authors to find a colleague to proofread or to use a reviewing service before the next submission

Dear referee

Thank you for taking your time to comment on this manuscript. First off, we are very sorry for the amount of technical errors in this manuscript! We have proofread the manuscript very carefully and hope that all the errors has been corrected. If the editor thinks that further proofreading could benefit the manuscript then we will have it proofread by a professional, when the final corrections has been added. We will address all the referee questions below.

2. In the paper, you are trying to combine several methods in a novel way to characterize the lake hydrology. Because of this, the methods must be clearer and not assume that the readers are familiar with the various components. The purpose of a certain method, the details of the models used, the interpretation of results needs to be more extensive. For example:

a. The PARAFAC analysis: How do you get?

PARAFAC modelling is a lengthy process involving many steps which is fully described in Murphy et al. (2013). We can add this to the manuscript if the editor believes it is of importance. To ease the understanding of PARAFAC modelling we have added a section regarding this to the introduction:

"Some non-conservative tracers such as fluorescent dissolved organic matter (FDOM), which can be determined using parallel factor analysis (PARAFAC), has been used to trace dissolved organic matter (DOM) in aquatic environments (He et al., 2014; Massicotte and Frenette, 2011; Stedmon et al., 2003; Stedmon and Markager, 2005b; Walker et al., 2009). PARAFAC analysis is a modelling tool which can separate multiple FDOM samples (emission and excitation matrices) into specific fluorescent components (Stedmon et al., 2003). The fluorescent components can be biological produced proteins derived from bacteria or molecules from the degradation of terrestrial organic material. These components has previously has been found visually using a single excitation emission matrix and the observed fluorescent peaks (Coble, 1996). The differentiation between the fluorescent components are both a strength and a weakness as we can isolate many different components, but all of them can differ in both degradation and production rate in the lake and groundwater. Furthermore, these FDOM components have not yet been investigated as tracers in groundwater fed lakes, as they, just as the rest of the non-conservative biological tracers, are volatile."

What is the interpretation of the results in Figure 3?

Figure 3 provides a visual depiction of the fluorescent components found in the PARAFAC modelling. The interpretation of the results is described in the section named "Fluorescent DOM" where we explain the origin of the components and if they are degradable or not.

To ease the understanding of figure 3 we have added the following sentence to the result section:

"PARAFAC and split–half analysis modelling identified five distinct fluorescent DOM components (C1-C5, explained variance 96.79 %). The spectral properties of the five fluorophores (components) identified by the PARAFAC analysis (Fig. 3) revealed that the DOM pool had both terrestrial and microbial influence."

b. The determination of WRT is unclear still. The text (Line 156) briefly describes that WRT was estimated through tracer concentrations if no degradation took place. There is no further mention beyond this paragraph and needs to be elaborated upon.

We have added the following to the introduction to describe "tracer concentrations if no degradation took place":

"This is observed as a change in tracer concentrations (often a decrease) after the groundwater is discharged to the lake. The speed of which the change in concentration occur are typically related to seasonal variations (e.g. temperature, mixing of the water column and UV-radiation) and the WRT of the lake e.g. the amount of time the tracer has been the lake. The removal and degradation rates have been examined in many instances e.g. for phosphorus (Larsen and Mercier, 1976; Vollenweider, 1975), nitrate (Harrison et al., 2009; Jensen et al., 1995), CDOM and DOC (Madsen-Østerbye et al., 2017). In a modelling approach these rates are important as they provide information about the change in tracer concentration, from the time when the tracer entered the lake. From this, it is possible to back-calculate the mixed inflow concentration of specific tracers when they were discharged to the lake. These estimations are crucial when working with non-conservative tracers, as it enables a direct comparison between the tracer concentration found in the catchment and the estimated mixed lake concentration before degradation took place."

Furthermore, we have also added information regarding the determination of WRT in the materials and method section:

"WRT of the lake were found using traditional hydrological methods combined with non-conservative tracer concentrations which were related to their degradation rates to form a proxy for the maximum WRT. Previous hydrological models suggested that the lake had a WRT between 0.4 and 3.3 years. To further narrow this range, we estimated the WRT by relating the concentrations found in the lake to their respective degradation rates related to increasing WRT e.g. by adding the estimated removed tracer since the groundwater entered the lake to the measured concentration in the lake. This enabled us to narrow the span of the WRT based on the estimated mixed inflowing tracer concentration related to the actual catchment concentrations. E.g. if the estimated inflow concentration of a tracer is 100 µmol l$^{-1}$, at a WRT of 2 years, and the highest catchment

tracer concentrations is 50 µmol l$^{-1}$ then the catchment do not support a WRT of 2 years."

3. I have some reservations on the approach taken with determining the groundwater discharge areas and lake WRT (section starting from line 215) or the methods have not been described adequately. I appreciate that this section was added but it is still not clear. When reading the methods, the reader needs to be able to see how you took your data and processed it, and arrive at the resulting figure or results. Additionally, I am not convinced that the equations used in this section were used correctly (see specific comments below).

Thank you for addressing this. We have added a flow diagram to the manuscript which we hope will improve the understanding of the different aspects in the material and method section. Furthermore, we have changed and streamlined the section so it now reads:

"In this instance, we estimated lake tracer concentrations of TN, TP, CDOM and DOC for WRTs from 0.25 to 3.5 years in 0.25 year increments following Eq. (1):

$$MIC = \frac{tr_{lake}}{ret\,(frac)},\tag{1}$$

where *MIC* is the mixed inflow concentration, $tr_{lake}$ is the tracer concentration found in the lake and *ret (frac)* is the retention fraction of the tracer at a known WRT. Retention models used in this study were based on the lake type as well as the geographical location of our lake. As there is not one model that can provide removal rates across all lakes we encourage the readers to find models related to their specific lake type. Thus, phosphorus equilibrium concentration in this study were found using Eq. (2) modified from Larsen and Mercier (1976) which describes phosphor retention in lakes with low productivity:

$$retP\,(frac) = 1 - \frac{1}{1+\sqrt{WRT}},\tag{2}$$

where *retP* (frac) is the retention fraction of phosphorus and *WRT* is the water retention time in the lake. Similarly, nitrate inflow concentration were estimated using a modified Danish nitrate removal model derived from Jensen et al. (1995) describing retention for shallow lakes with a short WRT (0-6 years)  Eq. (3):

$$retN\,(frac) = \frac{59 \cdot WRT^{0.29}}{100},\tag{3}$$

where *retN (frac)* is the retention fraction of nitrate and *WRT* is the water retention time in the lake. The corresponding retention fractions removed at different WRT were related to the lake concentrations to estimate what the mixed inflow concentration must have been to produce the present lake concentration. The combined summer UV-radiation and bacterial degradation rates of DOC and CDOM in groundwater from the dominating catchment vegetation type of the lake (Madsen-Østerbye et al., 2017) were extrapolated to the rest of the year. This was done by relating the degradation rates to the mean monthly UV-index (DMI, 2015) while assuming a linear relationship between the UV-index and degradation rates. Thus, enabling us to estimate the specific removal of DOC and CDOM on a monthly basis related to the concentration measured in the lake at the time of sampling following Eq. (4):

$$tr_{lake} = tr_{lakepm} - tr_{lakepm} \cdot degra\,(frac) - tr_{lakepm} \cdot mf + tr_{inflow} \cdot mff,\tag{4}$$

Where $tr_{lake}$ is the lake concentration in the specific month, $tr_{lakepm}$ is the lake tracer concentration in the previous month, *mff* is the monthly flushing fraction (mff = 1/WRT/12), *degra (frac)* is the degradation fraction

in present month related to UV-radiation and $tr_{inflow}$ is the inflowing tracer concentration. Eq. 4 was solved for $tr_{inflow}$ and calculated using the same WRTs as the nitrate and phosphorus models."

**Specific Comments**

Line 111 - The PARAFAC analysis was described as a three - way modelling tool but it is not clear between what three things

This sentence has been moved from the materials and methods section to the introduction and now reads:

"PARAFAC analysis is a modelling tool which can separate multiple FDOM samples (emission and excitation matrices) into specific fluorescent components (Stedmon et al., 2003). "

Line 149 – What are the lambda values here? You should briefly comment on what this is

This sentence has been added to the materials and method section:

"The model also computes lambda values from the least squares regression measuring which tracers are most influential on the relative fractions of water originating from the groundwater well sites. Lambda values therefore quantifies how much the relative contribution from the sites change when one tracer is changed a unit while the rest of the tracers are kept constant."

Line 153 – The text is describing the degradation of tracer concentrations in the previous sentence. Then it is followed by "This equilibrium estimations" … Is this referring to equilibrium concentrations? Degradation rates?

This sentence has been removed.

Line 165 – The Vollenweider equation that you provide as equation 2 is not the form provided in the 1975 paper you cite and has been used erroneously. From a mass balance approach, the mass fraction that is exported (C outflow /C initial) = 1/(1+k*WRT)

- If you have decided to replace the removal rate constant k with WRT^0.5, you need to justify this with further literature

- The general form of the equation that is presented in your paper looks to be the percent *expor*, rather than percent *retention.* You will have to use 1 - (1/(1+k*WRT) to get retention

- In fact, the form given in this manuscript shows that the percent retained increases with decreasing WRT.

Thank you for addressing this. Unfortunately, the wrong reference was added to the equation. This has been corrected and the equation has been modified to ease the reading of the material and methods section. The

"Retention models used in this study were based on the lake type as well as the geographical location of our lake. As there is not one model that can provide removal rates across all lakes we encourage the reader to find models related to the specific lake type. Thus, phosphorus equilibrium concentration in this study were found using Eq. (2) modified from Larsen and Mercier (1976) which describes phosphor retention in lakes with low productivity:

$$retP\ (frac) = 1 - \frac{1}{1+\sqrt{WRT}},$$ (2)

where *retP* (frac) is the retention fraction of phosphorus and *WRT* is the water retention time of the lake."

Line 169 – I find it difficult to see how equation 3 is applicable. The equation only applies for lakes that have a WRT of approximately 0 – 6.2 years (and I agree the study site fits inside this range). However, the source that was cite for this equation is in a Danish report published over twenty years ago and is not easily accessible. As an empirical equation, it is extremely difficult for the reader to understand the limitations, assumptions and the overall validity of this equation for this study (e.g. anyone trying to replicate your methods).

We have added further description regarding the choice of models used for N and P removal estimates. We believe that specific models used to described specific lakes should be chosen based on lake-type, WRT, catchment and climate. Following this we have added these sentences to the materials and methods section:

"Retention models used in this study were based on the lake type as well as the geographical location of our lake. As there is not one model that can provide removal rates across all lakes we encourage the reader to find models related to the specific lake type. Thus, phosphorus equilibrium concentration in this study were found using Eq. (2) modified from Larsen and Mercier (1976) which describes phosphor retention in lakes with low productivity."

"Similarly, nitrate inflow concentrations were estimated using a modified Danish nitrate removal model derived from Jensen et al. (1995) describing retention for shallow lakes with short WRT 0-6 years  Eq. (3):"

Line 178 – It is unclear if the authors developed equation 4 on their own or was from literature and thus should be clarified. Regardless, it appears that this equation is not internally consistent with respect to units. The monthly flushing rates are in units of 1/Time; thus the first two terms on the right hand side of the equation are in concentration units, whereas the last two term s are in concentration/time

We have changed the sentence to clarify that the monthly flushing units are not 1/time, but the fraction water removed from the lake each month. This enables the calculation of $tr_{lake}$ (the lake tracer concentration) on a monthly basis.

We have changed the sentence and have pointed out that the second part of the equation is not 1/time but the fraction of water flushed from the lake:

"Thus, enabling us to estimate the specific removal of DOC and CDOM on a monthly basis related to the concentration measured in the lake at the time of sampling following Eq. (4):

$$tr_{lake} = tr_{lakepm} - tr_{lakepm} \cdot degra\,(frac) - tr_{lakepm} \cdot mf + tr_{inflow} \cdot mff ,$$ (4)

Where $tr_{lake}$ is the lake concentration in the specific month, $tr_{lakepm}$ is the lake tracer concentration in the previous month, $mff$ is the monthly flushing fraction (mff = 1/WRT/12), $degra\,(frac)$ is the degradation fraction in present month related to UV-radiation and $tr_{inflow}$ is the inflowing tracer concentration. Eq. 4 was solved for $tr_{inflow}$ and calculated using the same WRTs as the nitrate and phosphorus models."

Line 180 – Is the peak degradation through UV - radiation determined linearly from the UV - time relationship? I.e. is the peak degradation of 100% assumed to coincide with the peak UV radiation?

There is a clear relationship between the seasons and degradation of CDOM and DOC (see "Photodegradation of DOC in a shallow prairie wetland: evidence from seasonal changes in DOC optical properties and chemical characteristics MARLEY J. WAISER* and RICHARD D. ROBARTS"). This seasonal change is related to the UV-radiation which is absorbed by DOC and CDOM in the top layer of the lake. In this study, we assume peak degradation with maximum UV-radiation.

Line 200 – "… combination of peaks N and T produced biological (Coble, 1996)." does not make sense.

What Ns and Ts are you talking about?

We have rewritten the sentence and added it to the introduction to explain that "peaks" are observed fluorescent peaks seen in a single excitation emission matrix. The sentence now reads:

"The fluorescent components can be biological produced proteins derived from bacteria or molecules from the degradation of terrestrial organic material. These components has previously has been found visually using a single excitation emission matrix and the observed fluorescent peaks (Coble, 1996)."

What biological is produced?

Peaks N and T are produced biologically. We have changed the following sentence to explain that the peaks are actually components that is produced biologically:

"Component C4 is similar to component 5 found in Stedmon et al. (2003) and is believed to be a combination of fluorescent labile materials named peak N and T which are produced biologically associated with DOM degradation (Coble, 1996; Stedmon and Markager, 2005b)."

Line 218 – The conclusion that the concentrations of TDN do not support a WRT value of over 2 years is wholly dependent on the model you choose. With the limited support provided for the model, this claim is not strong.

We have added further information regarding the nitrate model used as well as a note that models should be

based on the lake in question. The following sentences has been changed and added:

"Similarly, nitrate inflow concentration were estimated using a modified Danish nitrate removal model derived from Jensen et al. (1995) describing retention for shallow lakes with a short WRT (0-6 years)  Eq. (3):"

"Retention models used in this study were based on the lake type as well as the geographical location of our lake.  As there is not one model that can provide removal rates across all lakes we encourage the reader to find models related to the specific lake type.

Line 297 – I do not think the manuscript does an adequate job in convincing the reader that this method can capture uncommon or stochastic events. It did not seem like the samples taken were taken at different times of the year when extreme conditions occur. Is there literature that supports your claim that these environmental tracers capture these stochastic events? Even so, how would a single sampling campaign be used to extrapolate beyond the timeframe or snapshot of when you sampled?

We have changed the sentence to clarify what our intentions were:

"The multi-tracer approach enables the determination of discharge areas much more precisely and on a temporal scale related to the WRT of the lake (in this instance the previous 3 to 24 months). The model is therefore able to track uncommon events such as heavy precipitation where large amount of water with different tracer concentrations is discharged to the lake during a short period. These events are often difficult to track as seepage meters needs to be deployed in this period as well as in the right place."

Line 320 – I'm not sure that claiming the analysis remaining generally unchanged by running the CATS model with a 10% perturbation of tracer concentrations is sufficient. The tracer concentrations in your supplementary material show that TP, TN and DOC all can vary by an entire order of magnitude between sites. Can you be certain that they also cannot fluctuate by an order of magnitude throughout the year?

The differences in tracer concentration between sites are related to the catchment area and the percolating groundwater. We know the concentration within the groundwater wells fluctuates as well – often related to dry periods followed by rain events. We have seasonal measurements of the DOC and CDOM concentrations in a stationary groundwater well which, if the editor wants it, can be used as a proxy for changes in tracer concentrations. Samples taken after periods of drought followed by heavy rain can be removed from the dataset to calculate the yearly fluctuation in concentration which can be incorporated into the sensitivity analysis instead of a +-10 % change.

**Technical Corrections**

1. Line 17 – WRT has not been introduced yet, do not use abbreviation

This has been corrected

2. Line 17 – WRT was estimated to *be* 2 years

This has been corrected

3. Line 17 – Isolation of groundwater recharge areas *was*

This has been corrected

4. Line 18 - … sites with *a* high degree of recharge *were*

This has been corrected

5. Line 29 - … which to some *degree*

This has been corrected

6. Line 30 - … the groundwater contributes nutrients

This has been corrected

*7.* Line 37 - … *particularly* in small water bodies. *For example*

This has been corrected

8. Line 53 – 36 Cl

This has been corrected

9. Line 60 - You are talking about the nutrients (plural) so it should be "which *are* either remineralized when dying …". Also poorly phrased as nutrients (which is the subject of this sentence do not die)

This sentence has been completely rewritten

10. Line 66 – *fluorescent*

This has been corrected

11. Line 75 – use consistent style for lists throughout your paper. Either use (1) as you did in line 52 or keep to Also use a colon to introduce your lists, not a semicolon

This has been corrected throughout the manuscript

12. Line 84 – Subularia *aquatica*

The name has been changed to: "*Subularia aquatica*"

13. Line 90 – *preliminary* work

This has been corrected

14. Line 93 – within 5 - 45m of what?

The sentence has been changed to:

"A total of 30 groundwater samples were taken every 50 meters around the lake, within a distance of 5-45 m to the shore, in temporary groundwater wells at 1.25 meters of depth in February 2016"

15. Line 96 – *hermetically sealed*

This has been corrected

16. Line 100 – *quartz*

This has been corrected

17. Line 101 – δ18O is a ratio, not a concentration, please fix this throughout your paper

This has been corrected and we generally use $\delta^{18}O$ ‰ to describe the isotope throughout the manuscript. Furthermore, we have added a sentence to the introduction explaining how $\delta^{18}O$ is presented:

"Precipitation-derived environmental tracers, such as the isotope $\delta^{18}O$ (reported in the Vienna-standard mean ocean water (SMOW) where $\delta_{sample}$ ‰ = 1000(($R_{sample}/R_{smow}$)-1) and R is the $\delta^{18}O/\delta^{16}O$ ratio (Turner et al., 1987)), have been used to trace the interaction between ground and surface-water. As evaporation occurs in the surface water it becomes enriched with $\delta^{18}O$ producing a unique lake $\delta^{18}O/\delta^{16}O$ ratio which can be traced in the areas with groundwater recharge (Krabbenhoft et al., 1990)."

18. Line 107 – borate buffer *was*

This has been corrected

19. Line 117 – were subtracted *… to remove*

This has been corrected

20. Line 117 is a run on sentence; separate is at "the data were then Raman normalized …"

This has been corrected

21. Line 129 – *biologically inert*

This has been corrected

22. Line 134 is not a full sentence

This sentence has been complete rewritten

23. Line 137 – atmospheric *deposition*

This has been corrected

24. Line 139 – … linear in features. What features are you talking about?

This sentence and paragraph has been completely rewritten and added to the introduction:

"As the concentrations of both conservative and non-conservative tracers in a groundwater fed lake correspond to the mixed concentrations of discharging groundwater, while taking degradation and atmospheric deposition into account, it is possible to utilize the Community Assembly via Trait Selection approach (CATS). This model has been used to predict the relative abundances of a set of species from measures of community-aggregated trait values (average leaf area, root length etc.) for all plant species at a site (Shipley, 2010; Shipley et al., 2006, 2011). The CATS model has three main parameters: (1) it models the probabilities (2) that maximize the entropy (3) based on a set of constraints (Laliberté and Shipley, 2011; Shipley et al., 2011). In reality, the model (1) predicts the relative abundances of species at a location from their (3) average traits values by (2) minimizing the number of species that explain the mean traits values. The maximum entropy (2) is the maximizing of "new knowledge gained", related to plant communities this means that we are moving from "all species has the same relative abundances" to "a few species has a high relative abundance". When applying the model to the lake-groundwater interaction we use the measured tracer concentrations at groundwater well sites around the lake as the individual plant species and the estimated mixed lake concentration before degradation took place as the community-aggregated trait values."

25. Line 142 – What is the FD package?

It a R package which can be downloaded through the CRAN package repository. For more information please see https://cloud.r-project.org

26. Line 143 – In *the* present study

This has been corrected

27. Line 147 – The model outputs maximum entropy *probability* fractions

This has been corrected

---

## Author Comment (AC2) · 28 Dec 2017

Dear referee

Thank you for taking your time to comment on this manuscript. First off, we are very sorry for the amount of technical errors in this manuscript! We have proofread the manuscript very carefully and hope that all the errors has been corrected. If the editor thinks that further proofreading could benefit the manuscript then we will have it proofread by a professional, when the final corrections has been added. We will address all the referee questions below.

**Catchment tracers reveal discharge, recharge and sources of groundwater-borne pollutants in a novel lake modelling approach**

**General comments**

I will give some more general comments in the following part while some further specific and technical aspects are listed below this section.

1. The introduction might need a bit more structure. There is a lack of background information especially on the functioning of the tracer methods and especially the model applied. It would be good to introduce the different tracers and their functioning in detection of groundwater-lake interactions in more detail. While the investigation of oxygen isotopes might be familiar to a lot of readers, the functions and applicability of the different DOC/DOC fractions as a tracer for groundwater interactions with lakes might be helpful to know. This is especially important for the development of the hypothesis/research question where the FDOM fraction plays an important role. Also, the reader might need more information about the meaning of WRTs in this context (elaborate this in more detail in Line 65).

Furthermore, the CATS model as the novelty in the study because it is applied for the first time in such a context, is not mentioned in the introduction at all. I feel that the quality of the manuscript would improve quite a bit if the authors try to step back from their own view and reflect the structure and the information given in the introduction from the point of the potential reader. Elaborate in more details the links between the different tracers, the WRT, the CATS model etc. Also, give a brief and general description of the study design and the methods applied in order to allow the reader to go for the details in the Material and Methods-section following.

We agree that the introduction needs to describe the tracer methods and the model used in more detail. This, is why we've changed 2/3 of the introduction and added further information regarding FDOM, CATS, CDOM, DOC and nutrients, which is specified below.

"FDOM:

"Some non-conservative tracers such as fluorescent dissolved organic matter (FDOM), which can be determined using parallel factor analysis (PARAFAC), has been used to trace dissolved organic matter (DOM) in aquatic environments (He et al., 2014; Massicotte and Frenette, 2011; Stedmon et al., 2003; Stedmon and

Markager, 2005b; Walker et al., 2009). PARAFAC analysis is a modelling tool which can separate multiple FDOM samples (emission and excitation matrices) into specific fluorescent components (Stedmon et al., 2003). The fluorescent components can be biological produced proteins derived from bacteria or molecules from the degradation of terrestrial organic material. These components has previously has been found visually using a single excitation emission matrix and the observed fluorescent peaks (Coble, 1996). The differentiation between the fluorescent components are both a strength and a weakness as we can isolate many different components, but all of them can differ in both degradation and production rate in the lake and groundwater. Furthermore, these FDOM components have not yet been investigated as tracers in groundwater fed lakes, as they, just as the rest of the non-conservative biological tracers, are volatile."

CATS:

As the concentrations of both conservative and non-conservative tracers in a groundwater fed lake correspond to the mixed concentrations of discharging groundwater, while taking degradation and atmospheric deposition into account, it is possible to utilize the Community Assembly via Trait Selection approach (CATS). This model has been used to predict the relative abundances of a set of species from measures of community-aggregated trait values (average leaf area, root length etc.) for all plant species at a site (Shipley, 2010; Shipley et al., 2006, 2011). The CATS model has three main parameters: (1) it models the probabilities (2) that maximize the entropy (3) based on a set of constraints (Laliberté and Shipley, 2011; Shipley et al., 2011). In reality, the model (1) predicts the relative abundances of species at a location from their (3) average traits values by (2) minimizing the number of species that explain the mean traits values. The maximum entropy (2) is the maximizing of "new knowledge gained", related to plant communities this means that we are moving from "all species has the same relative abundances" to "a few species has a high relative abundance". When applying the model to the lake-groundwater interaction we use the measured tracer concentrations at groundwater well sites around the lake as the individual plant species and the estimated mixed lake concentration before degradation took place as the community-aggregated trait values.

CDOM, DOC and Nutrients:

This is observed as a change in tracer concentrations (often a decrease) after the groundwater is discharged to the lake. The speed of which the change in concentration happens are mostly related to seasonal variations (e.g. temperature, mixing of the water column and UV-radiation) and the WRT of the lake e.g. the amount of time the tracer is in the lake. The removal and degradation rates have been examined in many instances e.g. for phosphorus (Larsen and Mercier, 1976; Vollenweider, 1975), nitrate (Harrison et al., 2009; Jensen et al., 1995), CDOM and DOC (Madsen-Østerbye et al., 2017). In a modelling approach these rates are important as they provides information about the change in tracer concentration, from the time when the tracer entered the lake. From this it is possible to back-calculate what the mixed inflowing concentration of specific tracers were when they were discharged to the lake. These estimations are crucial when working with non-conservative tracers as it enables a direct comparison between the tracer concentration found in the catchment and the estimated mixed lake concentration before degradation took place."

2. Having said this, the authors start the method section by explaining groundwater sampling and sample/data analysis without introducing to the specific idea/concept/design of the study. This might be acceptable if the concept was already described in the introduction. However, as already mentioned above this is not the case which is why I strongly recommend adding the necessary information here.

We agree that the introduction needs to provide the specific idea, concept or design of the study. This is why we have added information regarding the CATS model (shown above) as well as information in the section, which describes the hypothesis of the study.

The section now reads:

"Determining groundwater movement using both conservative and non-conservative tracers found around the lake shore overcomes some fundamental shortcomings related traditional sampling. Firstly, we often measure tracers, which do not have a direct impact on the lake ecosystem and therefore do not provide meaningful information regarding the inflow of nutrients or CDOM. Furthermore, the sampling is only done in a few places throughout the catchment, which do not necessarily provide the all information regarding the groundwater flow patterns or to which degree water enters the lake and where. To overcome this, we measured conservative and relevant non-conservative tracers in and around a small lake with the aim of developing a novel approach to identify groundwater discharge and recharge areas on a high spatial scale. Thereby, pin-pointing areas which delivers pollutants to the lake, where groundwater recharge happens and where recharge occur with an increased flowrate of which the latter can spark further investigations into the lake WRT. Information regarding the WRT of the lake is especially useful when investigating how the concentrations of pollutants in the lake will develop after future restoration attempts. In the present study, we measured the eight following tracers: FDOM, CDOM, DOC, total dissolved phosphorus (TDP), total dissolved nitrogen (TDN), total phosphorus (TP), total nitrogen (TN) and $\delta^{18}O/\delta^{16}O$ isotope ratios and tested: (1) if groundwater discharge sites and pollutant sources can be estimated with the CATS model based on tracer concentrations, (2) if conservative and non-conservative tracers can be used to detect groundwater recharge areas as well provide insights into which areas have a high groundwater recharge rate and (3) if catchment-derived tracer concentrations can be used to estimate a range of WRTs which can be used with the CATS model. "

Since the application of the CATS model is a crucial part of the study it should be already described in the introduction.

We agree and have added it to the introduction (see above)

Also, since the reader until now is not aware of the concept of the study the headings "Groundwater recharge and areas of high recharge" (L132) is quite confusing since at first site such a differentiation does not make very much sense.

We agree and have changed the heading to:

**"Groundwater recharge and areas with a high groundwater recharge rate "**

Similar to that the heading "Groundwater discharge and lake WRT" (L135) makes the reader wonder why these two aspects are looked at together.

We agree and have changed the heading to:

**Non-conservative tracer degradation and lake WRT"**

Furthermore, we have included groundwater discharge in a separate section named "**The CATS model**"

3. About the CATS model:

  a.    The model identifies those sites around the lake which contribute most significantly to the lake´s tracer concentrations. From this, you conclude that at these sites most groundwater discharge takes place. However, if concentrations are high discharge volumes do not necessarily have to be large as well to influence the lake concentrations. For these kinds of approaches there are usually end-member-mixing models applied which I guess also the CATS model is of such a kind.

Yes, the CATS model resembles an end-member-mixing model. In this regard, the number of tracers used and their differentiation in concentrations are very important. Thus, higher number of tracers and higher un-correlated differences between the tracers results in a more secure determination of groundwater discharge sites.

The following has been added to the materials and methods section:

"When choosing tracers, it is important that there is differentiation between the concentrations measured at the sites. This implies that a higher number of tracers and higher un-correlated concentration differences between the sites results in a more reliable determination of groundwater discharge sites."

  b.    L147: What are "entropy probability fractions of the groundwater..."

We have added this section to the introduction and changed the sentence to make it clearer:

"The CATS model has three main parameters: (1) it models the probabilities (2) that maximize the entropy (3) based on a set of constraints (Laliberté and Shipley, 2011; Shipley et al., 2011). In reality, the model (1) predicts the relative abundances of species at a location from their (3) average traits values by (2) minimizing the number of species that explain the mean traits values. The maximum entropy (2) is the maximizing of "new knowledge gained", related to plant communities this means that we are moving from "all species has the same relative abundances" to "a few species has a high relative abundance"."

c. L150f: "The model also predicts lambda values from the least squares regression explaining which tracers are most influential on the relative fractions of water originating from the groundwater well sites."

→ do not fully understand what that means. To me this implies that depending on the tracer looked at the model output differs, i.e. if you look at P there are different sites relevant for lake concentrations than compared to CDOM. And to me that totally makes sense because the concentrations may spatially vary a lot within the same or among different parameters.

Regarding the lambda values. Lambda values are a measure of how much each tracer influence the model outcome. Meaning that, if all sites had the same tracer concentrations we would also have the same lambda values among tracers (which would be 0), but because we have differences in tracer concentration between sites we see a variation in lambda values. A higher lambda value means that the tracer varies a lot between sites and the sites varies from the estimated lake concentration – this means that a change in e.g. CDOM concentration would have a great impact on the model outcome.

We have added the following regarding lambda values to the introduction and the results section:

"The model also computes lambda values from the least squares regression measuring which tracers are most influential on the relative fractions of water originating from the groundwater well sites. Lambda values therefore quantifies how much the relative contribution from the sites change when one tracer is changed a unit while the rest of the tracers are kept constant. "

and

"Lambda values, explaining which tracers are the most important when predicting the fractions of water origination from groundwater well sites, showed that CDOM was the most important tracer when determining which sites delivered water to the lake with a mean lambda value for all WRTs of 24.2 versus 0.1-5.9 for the other tracers."

Regarding the part related to 3a, the model does not see the tracers as individuals at a site it rather looks at them as a "package" with each site having a distinct fingerprint of tracers. The sites delivering water to the lake are therefore determined by the combination of tracers they have and the differences amongst them makes the model stronger.

However, as already mentioned in 3a. the lake concentrations do not only depend on the concentrations but also on the volumes of groundwater discharge. But the discharge volume of a site is (to my understanding) completely independent from the parameter concentration which means that it is not possible to deduce

discharge volumes/portions from parameter concentrations. By that the model does not give you any reliable output of relevant discharge sites.

We have described how each site are seen as a "package" and the differences between the tracers makes the model able to estimate discharge fractions in the answer to 3a.

d.   How is the water retention time implemented in the CATS model? I guess the CATS model initially didn't have such kind of parameter to be included in the algorithm.

The CATS model was developed for terrestrial plant communities and does therefore not have any parameter regarding WRT. Instead WRT is incorporated into the community aggregated trait mean. Meaning, that to every WRT there is a certain mixed groundwater discharge tracer concentration if we are to measure the concentration actually found in the lake.

We have added the following to the introduction and materials and method section to clarify this:

"This is observed as a change in tracer concentrations (often a decrease) after the groundwater is discharged to the lake. The speed of which the change in concentration occurs are typically related to seasonal variations (e.g. temperature, mixing of the water column and UV-radiation) and the WRT of the lake e.g. the amount of time the tracer has been in the lake. The removal and degradation rates have been examined in many instances e.g. for phosphorus (Larsen and Mercier, 1976; Vollenweider, 1975), nitrate (Harrison et al., 2009; Jensen et al., 1995), CDOM and DOC (Madsen-Østerbye et al., 2017). In a modelling approach these rates are important as they provide information about the change in tracer concentration, from the time when the tracer entered the lake. From this, it is possible to back-calculate the mixed inflow concentration of specific tracers when they were discharged to the lake. These estimations are crucial when working with non-conservative tracers, as it enables a direct comparison between the tracer concentration found in the catchment and the estimated mixed lake concentration before degradation took place."

"WRT of the lake were found using traditional hydrological methods combined with non-conservative tracer concentrations which were related to their degradation rates to form a proxy for the maximum WRT. Previous hydrological models suggested that the lake had a WRT between 0.4 and 3.3 years. To further narrow this range, we estimated the WRT by relating the concentrations found in the lake to their respective degradation rates related to increasing WRT e.g. by adding the estimated removed tracer since the groundwater entered the lake to the measured concentration in the lake. This enabled us to narrow the span of the WRT based on the estimated mixed inflowing tracer concentration related to the actual catchment concentrations. E.g. if the estimated inflow concentration of a tracer is 100 μmol l$^{-1}$, at a WRT of 2 years, and the highest catchment tracer concentrations is 50 μmol l$^{-1}$ then the catchment do not support a WRT of 2 years."

About the results section: Provide more evidence for what you describe in the text by figures, or tables.

We have added a flow diagram to ease the reading of the materials and method section as well as the results section. Furthermore, we can add the estimations of the mixed inflowing tracer concentrations at different WRTs if the editor wishes that?

4. About the discussion: You start the section by phrasing three questions which are not all corresponding to the research questions phrased in the introduction. Try to be consistent here.

The 1$^{st}$ question was referring to the main questions from the introduction while the 2$^{nd}$ and 3$^{rd}$ question were based on a more general discussion of the results.

To clarify this, we have changed the sentence so it now reads.

"Based on the model results and earlier hydrological studies we will discuss the main questions from the introduction (1) if groundwater discharge sites and pollutant sources can be estimated with the CATS model based on tracer concentrations, (2) if conservative and non-conservative tracers can be used to detect groundwater recharge areas as well provide insights into which areas have a high groundwater recharge rate and (3) if catchment-derived tracer concentrations can be used to estimate a range of the water retention times which can be used with the CATS model. Furthermore, we will discuss which of the tracers works and which could possibly work with refined methods? And how these findings could benefit lake restoration programs."

**Specific comments**

L14 Within 5-45 m distance (?) to the shore

Yes. This has been corrected and now reads:

"installed every 50 m within a distance of 5-45 m to the shore"

L17 What is WRT?

This has been corrected and now reads:

"water retention times (WRTs)"

L30 The sentence implies that nutrients such as P, N, and C are contaminants which is not necessarily true. See also Line 10 in the abstract.

We agree that this is not always the case, at all, we hope it is clear as we write:

"Particularly in smaller lakes and ponds the groundwater contributes nutrients, dissolved organic carbon (DOC), colored dissolved organic matter (CDOM) or other contaminants, which can have a negative impact on the

biological quality of lakes"

Stating that it can have a negative impact.

L45 The water does not leave the lake bottom but the lake via the lake bottom.

This has been corrected

L52 about environmental tracers: Besides atmospheric tracers: What about lithospheric or pedosphere tracers?

The sentence has been changed to include lithospheric or pedospheric tracers from the catchment:

"These tracers are divided into three main categories: (1) environmental tracers (natural derived tracers from the atmosphere or catchment which are transported to the system)"

L58 I do not understand the last part of the sentence: What does "groundwater of different ages and origins" mean in the context of percolating groundwater?

"groundwater of different ages and origins" refers to that different catchments and flow rates influence the chemical levels in the groundwater.

The sentence has been changed to:

"Therefore, we propose a different approach utilizing both conservative and non-conservative tracers such as dissolved organic carbon and nutrients which are partly transferred to the percolating groundwater on its way to the lake (Kidmose et al., 2011). "

L60ff In this paragraph you are talking about the fate of inflowing components. However, (although for many readers probably quite obvious) you do not explain why this is important for your study.

The whole section has been rewritten and should now clearly state why the fate of the inflowing tracers is important for this study. The section now reads:

"This is observed as a change in tracer concentrations (often a decrease) after the groundwater is discharged to the lake. The speed of which the change in concentration occurs are typically related to seasonal variations (e.g. temperature, mixing of the water column and UV-radiation) and the WRT of the lake e.g. the amount of time the tracer is in the lake. The removal and degradation rates have been examined in many instances e.g. for phosphorus (Larsen and Mercier, 1976; Vollenweider, 1975), nitrate (Harrison et al., 2009; Jensen et al., 1995), CDOM and DOC (Madsen-Østerbye et al., 2017). In a modelling approach these rates are important as they provide information about the change in tracer concentration, from the time when the tracer entered the lake. From this, it is possible to back-calculate the mixed inflowing concentration of specific tracers when they were discharged to the lake. These estimations are crucial when working with non-conservative tracers, as it

enables a direct comparison between the tracer concentration found in the catchment and the estimated mixed lake concentration before degradation took place."

L74 Measurements were done not only around the lake but also in the lake, i. e. of the lake water, right?

Yes, this has been corrected

L74f Why are the existing approaches/tracers not sufficient? Maybe add something like "... to overcome...." to the first sentence of that paragraph.

We have added the following to describe why existing approaches and tracers are now sufficient:

"Determining groundwater movement using both conservative and non-conservative tracers found around along the lake shore overcomes some fundamental shortcomings related traditional sampling. Firstly, we often measure tracers which do not have a direct impact on the lake ecosystem and therefore do not provide meaningful information regarding the inflow of nutrients or CDOM. Furthermore, the sampling is only done in a few places throughout the catchment which do not necessarily provide the all information regarding the groundwater flow patterns or to which degree water enters the lake and where. To overcome this, we measured conservative and relevant non-conservative tracers in and around a small lake with the aim of developing a novel approach to identify groundwater discharge and recharge areas on a high spatial scale. Thereby, pin-pointing areas which delivers pollutants to the lake, where groundwater recharge happens and where recharge occur with an increased flowrate of which the latter can spark further investigations into the lake WRT. Information regarding the WRT of the lake is especially useful when investigating how the concentrations of pollutants in the lake will develop after future restoration attempts."

What methods/tracers do you actually combine and develop towards a new approach? Be more precise and at least list them here.

The section now reads:

"In the present study, we measured the eight following tracers: FDOM, CDOM, DOC, total dissolved phosphorus (TDP), total dissolved nitrogen (TDN), total phosphorus (TP), total nitrogen (TN) and $\delta^{18}O/\delta^{16}O$ isotope ratios"

L75ff Here you come up with a maximum entropy model which was not mentioned before. Is that the novel approach you are talking about before? So far, the reader might have considered the application of the tracers more or less described above to be the novel approach in this study.

Information on the maximum entropy and CATS model has now been added to the introduction and can be seen in the answer to question 1.

L77f What would be the benefit of a combined investigation of conservative and non-conservative tracers? And

again: Which tracers will you use for this investigation?

We have added this to the introduction which can be seen in the answer to L74f.

L78f Why are you interested in WRT? Where is the connection to the groundwater-lake interactions?

We have changed the sentence and added further information to the introduction to explain why WRT is of importance:

"(3) if catchment-derived tracer concentrations can be used to estimate a range of the water retention times which can be used with the CATS model. "

"Information regarding the WRT of the lake is especially useful when investigating how the concentrations of pollutants in the lake will develop after future restoration attempts."

L86 "to bypass water": Is that surface runoff that is collected in the drainage channel?

Before 1992 there were at stream going through the lake. We have specified that in the sentence:

"This led to a restoration attempt in 1992 where a channel was established to bypass the stream going through the lake, thus making the lake groundwater fed."

L93 Does "5-45 m" relate to the distance to the shore?

Yes. We have adjusted the sentence:

"A total of 30 groundwater samples were taken every 50 meters around the lake, within a distance of 5-45 m to the shore, in temporary groundwater wells at 1.25 meters of depth in February 2016."

L111 What exactly are the three steps of PARAFAC modelling?

The three-way modelling is done on the basis of (1) multiple (2) emission and (3) excitation FDOM signals. Information regarding PARAFAC has now been added to the introduction. Specifically, this sentence:

"PARAFAC analysis is a modelling tool which can separate multiple FDOM samples (emission and excitation matrices) into specific fluorescent components (Stedmon et al., 2003)"

L111  What information do the "specific fluorescent components" provide? Be more specific here (and maybe introduce this background information on FDOM already in the introduction)

Information regarding PARAFAC and FDOM has been added to the introduction including this sentence:

"The fluorescent components can be proteins derived from bacteria or molecules from the degradation of

terrestrial organic material. These components has previously been found visually using single excitation emission matrixes and their observed fluorescent peaks (Coble, 1996)."

L112  What is the "inner filter effect"?

An explanation of inner filter effect has been added to the materials and method section:

"The FDOM samples were initially diluted 2-12 times to account for self-quenching, also referred to as inner filter effect, which occurs with high CDOM absorbance in the sample (Kothawala et al., 2013)."

L113  "...which were measured spectrophotometrically ..."does this refer to CDOM or FDOM and if it refers to CDOM it implies that the samples from each site were measured twice for CDOM (see L99f). Why was that?

The sentence has been removed as this measurement of absorbance were only done to determine the dilution factor of the FDOM samples to correct for inner filter effect.  This is done by measuring the total absorbance (the sum of absorbance at excitation and emission wavelengths) which should not be above 0.042.

L117 How can the data be "Raman normalized" when the "Raman scattering" was "removed" from the data before?

This sentence has been changed in the materials and methods section to explain how the normalization took place:

"The data were then Raman normalized by dividing the florescent intensities by the integral of the Raman peak of the blank sample (Lawaetz and Stedmon, 2009)"

L129f "Changes in ...." This sentence was probably supposed to explain the concept of using oxygen isotopes as tracers for groundwater-lake interaction but in fact it does not. The authors should rework and complement the description of the use of O2 isotopes (again, this general introduction of the approach should probably go to the introduction section of the manuscript).

The following section has been added to the introduction to account for isotope utilization as a tracer for groundwater movement and determination of WRT.

"Precipitation-derived environmental tracers, such as the isotope $\delta^{18}O$ (reported in the Vienna-standard mean ocean water (SMOW) where $\delta_{sample}$ ‰ = 1000(($R_{sample}/R_{smow}$)-1) and R is the $\delta^{18}O/\delta^{16}O$ ratio (Turner et al., 1987)), have been used to trace the interaction between ground and surface-water. As evaporation occurs in the surface water it becomes enriched with $\delta^{18}O$ producing a unique lake $\delta^{18}O/\delta^{16}O$ ratio which can be traced in the areas with groundwater recharge (Krabbenhoft et al., 1990). The isotopic composition can be also be related to evaporation lines (from the $\delta^{18}O$ relationship $\delta^2H$) to estimate WRT (Gates et al., 2008; Gibson et al., 2002)."

L130 "However, deviations from lake 18O concentration was not observed in areas with groundwater recharge due a sampling depth of 1.25 m close to the lake." do not understand this sentence. 18O values in groundwater similar to lake water values indicate lake water infiltration into the aquifer which is why the sentence to me seems to be a circular reference. Also, I am not sure if the last part of the sentence refers to the sampling depth or the distance to the lake shore.

We have removed this sentence which was previously added to explain why we chose $\delta^{18}$O as a tracer for groundwater discharge. And added that to the introduction:

"As evaporation occurs in the surface water it becomes enriched with $\delta^{18}$O producing a unique lake $\delta^{18}$O/$\delta^{16}$O ratio which can be traced in the areas with groundwater recharge (Krabbenhoft et al., 1990)."

L131 "Groundwater well sites which clustered with the lake were considered as being groundwater recharge sites and were removed for the later estimations of groundwater discharge sites." What does this sentence mean?

We have clarified this in the following sentence and have added a flow diagram which illustrates the work flow used in this study:

"Groundwater well sites which formed a cluster together with the lake samples were considered as being groundwater recharge sites, e.g. water originating from the lake, and was excluded for the later estimations of groundwater discharge sites. The groundwater recharge sites were further investigated using a range non-conservative tracers influenced by biological degradation."

L132f If I understand the meaning of the sentence and its link to the o2 isotope section above correctly I would recommend modifying it as follows:

"Hierarchical Euclidean clustering was also done for the fluorescence components from the PARAFAC modelling."

To clarify the use if fluorescent components in a Hierarchical Euclidean cluster we have added the following to the material and methods section:

"The groundwater recharge sites were further investigated using a range of non-conservative tracers influenced by biological degradation. We found that some of the tracer concentrations changed when going from the lake to the groundwater. For example, CDOM which showed a decrease in concentration when entering the groundwater properly due to pH changes in the soil. An inspection of the results revealed that a protein based fluorescent component met our criteria of being (1) non-conservative, (2) not afflicted by the lake-groundwater interface and (3) not to easily degraded or produced in high amounts which could create false positive groundwater recharge sites. The PARAFAC component was related to the lake concentration with a Hierarchical Euclidean cluster dendrogram and the sites which clustered together with the lake samples indicated a high groundwater recharge rate."

L133 This is not a complete sentence. It does not have a verb. I assume that the authors want to express that they applied two different tracers in the same way in order to validate their findings.

We have completely rewritten this part, please see the section above.

L136ff This applies only in exclusively groundwater-fed lakes which should be indicated in this sentence.

This part has been rewritten

L138ff "Probabilities" of what?

What are the "constraints" and the "linear features"? If possible try to explain these very abstract terms and relations a bit more. If this is not possible I would consider removing this sentence.

We have added a section regarding the CATS model in the introduction and made it much simpler:

"As the concentrations of both conservative and non-conservative tracers in a groundwater fed lake correspond to the mixed concentrations of discharging groundwater, while taking degradation and atmospheric deposition into account, it is possible to utilize the Community Assembly via Trait Selection approach (CATS). This model has been used to predict the relative abundances of a set of species from measures of community-aggregated trait values (average leaf area, root length etc.) for all plant species at a site (Shipley, 2010; Shipley et al., 2006, 2011). The CATS model has three main parameters: (1) it models the probabilities (2) that maximize the entropy (3) based on a set of constraints (Laliberté and Shipley, 2011; Shipley et al., 2011). In reality, the model (1) predicts the relative abundances of species at a location from their (3) average traits values by (2) minimizing the number of species that explain the mean traits values. The maximum entropy (2) is the maximizing of "new knowledge gained", related to plant communities this means that we are moving from "all species has the same relative abundances" to "a few species has a high relative abundance". When applying the model to the lake-groundwater interaction we use the measured tracer concentrations at groundwater well sites around the lake as the individual plant species and the estimated mixed lake concentration before degradation took place as the community-aggregated trait values."

L140f I assume that "species" are "plant species"?

Yes, the sentence now says "plant species"

L142 At least give information on the meaning of the abbreviation of the R package mentioned.

This has been added: "FD (functional diversity)"

L143ff For which parameters/tracers have you run the CATS model? deltaO2, N, P, CDOM, FDOM, DOC? And

was the model run for each parameter separately or for all parameters simultaneously? In the first case: Would you not have to go for a more detailed discussion of the outcomes of the different model runs?

All parameters are incorporated in the model as a distinct fingerprint or package for each site. We have completely changed the section which now reads:

"In the present study, the concentrations of non-conservative tracers (DOC, CDOM, TDP and TDN) at groundwater well sites around the lake acted as the individual plant species at a site and the equilibrium tracer concentrations derived from Eq. 1 (DOC, CDOM, TP and TN) acted as the community-aggregated trait values. When choosing tracers, it is important that there is differentiation between the concentrations measured at the sites. Meaning that a higher number of tracers and higher un-correlated concentration differences between the sites results in a more secure determination of groundwater discharge sites. All tracers were investigated as a combined package, e.g. a single site are described by all the tracers mentioned above, and was run using the maxent function in the FD (functional diversity) package in R to compute the CATS model (Laliberté and Shipley, 2011)."

L145ff Isn´t it more like the model identifies those sites along the lake which contribute most to the lake?

This has been clarified in the sentence:

"From this, the model predicts the minimum number of groundwater well sites along the lake shore that explains the measured concentrations in the recipient lake by maximizing the sites relative contribution."

L153 Is "equilibrium" the tracer concentration in the lake which would be the same as MIC in Eq. 1?

Yes, this has been clarified in the material and methods section and in the result section:

"In the present study, the concentrations of non-conservative tracers (DOC, CDOM, TDP and TDN) at groundwater well sites around the lake acted as the individual plant species at a site and the equilibrium tracer concentrations derived from Eq. 1 (DOC, CDOM, TP and TN) acted as the community-aggregated trait values."

And

"Equilibrium tracer concentrations of DOC, CDOM, TDP and TDN (found using Eq. 1-4) for water retention times between 0.25 and 3.5 years in 0.25 increments revealed that concentrations of TDN in the catchment are not high enough to support WRT-values over 2 years based on the nitrogen retention model used."

L173ff "The combined summer UV-radiation and bacterial degradation rates of DOC and CDOM in groundwater from the dominating catchment vegetation type of the lake (Madsen Østerbye et al., 2017) were extrapolated to the rest of the year."

    a.     What does the vegetation type of the lake have to do with UV- and microbial degradation of C-fractions?

The vegetation type in the catchment of the lake matters as the degradation of CDOM and DOC are influenced by the molecular structure. The molecular structure change depending of the vegetation in the catchment (e.g. heathland or coniferous forest)

      b.      Why was it necessary to extrapolate to the rest of the year? To get the specific removal rates for month of the sampling?

UV-radiation acts as a photo catalyst for the degradation of CDOM and DOC. Thus, degradation is correlated to the amount of UV-radiation which varies yearly (high levels during summer and low levels during winter). To estimate the degradation in the lake during the past months/years we therefore have to know how much UV-radiation there has been during the months of degradation.

L174f "This was done by relating the rates to the mean monthly UV index (DMI, 2015) while assuming a linear relationship between the UV-index and degradation rates." is there are literature reference which proves that this a legitimate way of doing this?

Almost all UV-absorption in the lake is done by CDOM and DOC in the top 5 cm of the water column. CDOM concentrations are therefore highly correlated to the yearly UV-radiation. The degradation follows a seasonal pattern described in ""Photodegradation of DOC in a shallow prairie wetland: evidence from seasonal changes in DOC optical properties and chemical characteristics MARLEY J. WAISER* and RICHARD D. ROBARTS".

L182f "Eq. 4 was solved in relation to $tr_{inflow}$ and calculated using the same WRTs as the nitrate and phosphorus models."do you mean "... was solved for $tr_{inflow}$..."?

Yes, this has been corrected.

L191 The split-half analysis modelling has never been mentioned before. What is that and what does it do?

We have added a sentence regarding this in the materials and methods section which explains the role of the "split-half" analysis:

"A split–half analysis, were the dataset is split into two parts and compared multiple times, were used to test the results found in the PARAFAC model."

L192 "Component C1 was similar to previously found humic-like material..." does this relate to material of this study site? Which one? Terrestrial, aquatic?

The component, a molecular structure dissolved in the sample water, here named component C1 has been found before. By comparing the specific excitation-emission signal shown in fig. 3 to other studies we are able classify it as a component that most likely originate from the degradation of terrestrial matter (leaves etc.). The component is washed into the groundwater and further into the lake from the soil.

L192ff "The component absorbs in the UV-C region which has low intensities at the ground surface (Diffey, 2002) and are is therefore expected to be photo-resistant (Ishii and Boyer, 2012)."

    a.    Is "ground surface" = lake bottom?

    b.    I am not sure why you can conclude that the material is photo-resistant from the fact that UV-C has low intensities at the lake bottom.

By ground surface we mean the surface of the earth. We have changed the sentence:

"The component absorbs in the UV-C region which is absorbed by the ozone layer and atmosphere (Diffey, 2002) and is therefore expected to be mainly photo-resistant (Ishii and Boyer, 2012)."

L199 "Component C3 may be an intermediate product since concentration changes even in open oceans"

    a.    Intermediate product from what?
    b.    What do changes in concentrations in the open ocean imply?

We have changed the sentence to address your question:

"Component C3 may be an intermediate product or produced biologically since changes in the concentration have been observed in the open oceans and in sea ice which has no apparent connection to the terrestrial environment (Ishii and Boyer, 2012)."

L200 "Component C4 was found to be similar to one in Stedmon et al. (2003) and are is believed to be a combination of peaks N and T produced biologically..."

    a.    Please explain "peaks N and T produced biological". Besides the fact that the sentence doesn´t seem to make sense grammatically it is also not clear what the peaks are and why they are able to biologically produce an FDOM component. Furthermore, T is not explained.

The term "peaks" are widely used when examining fluorescence. Before utilizing PARAFAC modelling researchers would often use "peak peeking" to describe fluorescent peaks. The total fluorescent signal of a sample is a combination of many peaks, but only with PARAFAC are we able to isolate the different peaks which we then name "components". Coble (1996) isolated some of these peaks and named them, for example peak N and T.

We have changed the sentence to:

"Component C4 is similar to component 5 found in Stedmon et al. (2003) and is believed to be a combination of fluorescent labile materials named peak N and T which are produced biologically associated with DOM degradation (Coble, 1996; Stedmon and Markager, 2005b)."

L211f "Component C4 was chosen as a proxy for groundwater recharge as the concentration of the C4 component increase with biological activity and time in the groundwater."

    a.   How do you know that the concentration of C4 increases with biological activity and time in the groundwater? Since you have neither investigated biological activity nor groundwater residence times this information has to be from some literature. Please cite.

We have changed a sentence to clarify that component C4 is produced biologically. Furthermore, the degradation is related to the time spend in the soil/groundwater were biological activity are higher due to the increased surface area for bacteria.

"Component C4 is similar to component 5 found in Stedmon et al. (2003) and is believed to be a combination of fluorescent labile materials named peak N and T which are produced biologically associated with DOM degradation (Coble, 1996; Stedmon and Markager, 2005b)."

    b.   I do not fully understand the conclusion, i.e. why you can conclude that C4 is suitable as a groundwater tracer because its concentrations increase with biological activity? Increased biological activities can be found in many environments and at very small scales (e.g., also in the lake sediments).

We found that some tracers changed very much when going from the lake to the groundwater. For example, CDOM which showed a decrease in concentration when entering the groundwater properly due to pH changes in the soil. An inspection of the results revealed that a protein based fluorescent component met our criteria of being (1) non-conservative, (2) not afflicted by the lake-groundwater interface and (3) not to easily degraded or produced in high amounts, which could create false positive groundwater recharge sites. The sediments in the areas are sandy, as stated in the introduction, and have been washed by the recharging groundwater for many years. This leads us to believe that the biological activity does not differ by a large margin between the sites. We have added the following to the materials and method section regarding the choice of tracer when determining groundwater recharge:

"The groundwater recharge sites were further investigated using a range of non-conservative tracers influenced by biological degradation. We found that some of the tracer concentrations changed when going from the lake to the groundwater. For example, CDOM which showed a decrease in concentration when entering the groundwater properly due to pH changes in the soil. An inspection of the results revealed that a protein based fluorescent component met our criteria of being (1) non-conservative, (2) not afflicted by the lake-groundwater interface and (3) not to easily degraded or produced in high amounts which could create false positive groundwater recharge sites. The PARAFAC component was related to the lake concentration with a Hierarchical Euclidean cluster dendrogram and the sites which clustered together with the lake samples indicated a high groundwater recharge rate."

L216 "Tracer concentrations of the lake water narrowed down the possible WRT of the lake."

This has been corrected

L216ff "Equilibrium tracer concentrations of DOC, CDOM, TDP and TDN for water retention times between 0.25 and 3.5 years in 0.25 increments revealed that concentrations of TDN in the catchment are not high enough to support WRT-values over 2 years."

Does that mean that you calculated WRTs with each of the parameters individually and then picked the one most plausible? What is the range of results you got from those calculations? Show in table or graph and discuss why the same results differ (they shouldn´t differ very much, right?).

We have investigated tracer concentrations in the lake and from these we calculated the mixed inflowing concentration if no degradation took place. This mixed groundwater discharge concentration increases as WRT increases to a point where no concentration in the catchment can match the estimated mixed concentration. The ranges of the estimated concentrations are described in the result section, but we can provide the estimated concentrations as a table if the editor feels that it will contribute to the manuscript?

L220f "Groundwater discharge areas were found using the CATS model combined with applied to (?) nutrient concentrations and dissolved organic matter fractions estimated in relation to WRTs between 0.25 and 2 years."

We have changed the sentence:

"Groundwater discharge areas were found using the CATS model using nutrient concentrations and dissolved organic matter fractions estimated in with Eq. 1. related to WRTs between 0.25 and 2 years."

L221f "The estimated phosphorus concentrations ranged from 46 to 80 µg P l-1..."

   a.    Does that refer to MIC in Eq. 1?

Yes and Eq. 2 which needs to be combined to estimate the mixed inflow concentration.

The following has been added to the sentences regarding P, N, CDOM and DOC:

"The estimated phosphorus concentrations ranged from 46 to 80 µg P $l^{-1}$ (Eq. 1 and Eq. 2) while nitrate concentrations ranged from 1113 to 2417 µg N $l^{-1}$ (Eq. 1 and Eq. 3)."

and

"Thus, estimated mixed inflow concentrations of CDOM ranged from $A_{CDOM}$(340) = 0.43 to 1.04 $cm^{-1}$ and DOC ranged from 1205 µmol $l^{-1}$ to 3160 µmol $l^{-1}$ for WRT between 0.25 and 2 years (Eq. 1 and Eq. 4). "

b.   I assume the given concentration range refers to the time increments for which you run the model. Is that correct?

Yes, that is absolutely correct

L224 Explain $A_{CDOM}(340)$ and be consistent in using the same way/units for this parameter throughout the whole manuscript.

We have checked for consistency throughout the manuscript and corrected the mistakes. We have changed this sentence in the materials and method section to explain $A_{CDOM}(340)$ cm$^{-1}$:

"The CDOM absorbance was measured on a spectrophotometer (UV-1800, SHIMADZU, Japan) between 240 and 750 nm in 1 nm intervals in a 1 cm quartz glass cuvette and expressed as the absorbance at 340 nm ($A_{CDOM}(340)$ cm$^{-1}$)."

L228f  "The model identified the sites 1, 9, 11, 13 and 14 as the possible groundwater discharge sites for all WRTs (Fig. 5)."Figure 5 does not show that. It shows modelling results for the sites mentioned but this does not show the results of the other sites.

The word "possible" has been removed and the sentence has been reworked to avoid any misunderstandings. We have also added information of the cut-off point for the sites we don't include:

"The model identified the sites 1, 9, 11, 13 and 14 as the groundwater discharge sites delivering more than 0.1 % of the water for all WRTs (Fig. 5). Changes in site distribution and fractions of discharging water were observed between the different WRTs, but in general groundwater from 3-4 sites out of xx explain the estimated concentrations in the lake (Fig. 5)."

L236ff Add table or figure with lambda values

We have added the range of the lambda values for CDOM to the text:

"Lambda values, explaining which tracers are the most important when predicting the fractions of water origination from groundwater well sites, showed that CDOM was the most important tracer when determining which sites delivered water to the lake with a mean lambda value for all WRTs of 24.2 versus 0.1-5.9 for the other tracers."

L251  "... thus facilitating.... " does that refer to the drainage channels? Should be "which facilitate ..." or something?

Yes, it refers to the drainage channels. The sentence has been edited:

"The exact same areas are also the ones with adjacent drainage channels (Fig. 1a), which facilitate the areas as

recharge sites."

L254ff "Sites resembling the fluorescence found in the lake will indicate flowing water, while a difference in components between lake and groundwater sites will indicate a lower flow rate where there is sufficient time for a significant modification of the components representing the DOM pool."

   a. What is meant by flowing water? What is meant by differing between "flowing water" and "sites with lower flow rates"? Please be more specific and refer to the scientific correct terms.
   b. Which DOM pool do you refer to hear? Be more specific to facilitate easy understanding by the reader.

The sentence has been changed and flow rate is now termed groundwater recharge rate we have also added information to help the understanding regarding where the DOM pool originates from:

"Sites resembling the fluorescence found in the lake will indicate a high groundwater recharge rate, while a difference in components between lake and groundwater sites will indicate a lower groundwater recharge rate where there is sufficient time for a significant modification of the components representing the DOM pool of the lake."

L257ff I still do not necessarily see the link between microbial activity and the use of this component as indicator for groundwater discharge since microbial activity is not at all a process restricted to groundwater or aquifers.

If the editor thinks that this will improve the manuscript we will add a section regarding higher microbial degradation lake sediments than in lake water?

L264f "CDOM generally showed much lower absorbance at groundwater recharge sites than in the lake making it less suitable for estimating recharge areas. "Maybe I missed it but I haven´t found this information in the results section.

This information can be found in the supplementary and can be added to the manuscript if the editor finds it suited?

L267 "While component C1 was not particularly useful for estimating groundwater recharge, it could potentially be useful to estimate discharge sites."

a. This sentence sounds very vague (a lot of "could" "potentially" "estimate"). Consider rephrasing.

b. I do not follow the argumentation that C1 is a discharge indicator. Isn´t photo-resistance irrelevant in groundwater environments?

c. Also, when the component is photo-resistant and resistant to microbial degradation which other factors lead

to a degradation which can be related to increased groundwater discharge at sites 9 and 11?

The sentence has been changed to improve the readability:

"While component C1 was not particularly useful for estimating groundwater recharge, it could be useful to estimate discharge sites. To utilize the component for discharge estimates there is a need for an assessment of the degradation rate. While it has been shown that component C1 is largely photo-resistant, as it does not absorb in the UV-A radiation areas, and is largely resistant to microbial degradation processes (Ishii and Boyer, 2012) no reliable rates for the degradation has been found. In this study, we found that only sites number 9 and number 11 hold concentrations lower than the lake (Table S1) indicating that most groundwater discharge would originate from these sites if little to no degradation takes place."

L282 "...hinting that the lake is influenced by other water sources." What other sources could that be since the lake is solely groundwater fed? Besides atmospheric deposition there shouldn't be any other options.

We meant other sources than the southern bank. The sentence has been edited to ease the readers understanding.

"This does not correspond to tracer concentrations found in the southern area which show very high CDOM absorbance at 340 nm ($A_{CDOM}$(340) = 1.3-3.1 cm$^{-1}$) and DOC concentrations (3114-10467 µmol l$^{-1}$) in relation to the lake ($A_{CDOM}$(340) = 0.4 cm$^{-1}$/DOC 1058 µmol l$^{-1}$) hinting that the lake is influenced by groundwater discharge from other areas as well."

L289f "Seepage meter measurements from this area showed both discharging and recharging of groundwater (Solvang, 2016). "Which area are you referring to? From the sentence before it could be any.

The sentence has been changed to illustrate that we're talking about the eastern shore:

"Seepage meter measurements from the eastern shore showed both discharging and recharging of groundwater (Solvang, 2016)."

L291f "...indicating an influence of newly precipitated water or discharge and recharge of groundwater." Do the results indicate discharge or recharge? Indicating both at the same time is not possible.

We have changed the sentence so it now reads:

"The same was observed for $\delta^{18}$O samples from the eastern part of the lake, which were lower than the southern groundwater, indicating an influence of newly precipitated water or influence from the lake."

L298 Please elaborate how your approach is able to capture the dynamics you describe when you have only a

single sampling date.

When investigating groundwater discharge we often sample during short periods (with seepage meters for example) or during longer periods using hydraulic head (where we still have to measure the depth of the groundwater and the sampling points are often sparse and far apart). We're able to trace these events as we look into the lake tracer concentrations which is influenced by events taking place the last 0-2 years and by sampling at a representative time of the year we make sure that the groundwater tracer concentrations are representative.

We have changed the sentence to clarify:

"The model is therefore able to track uncommon events such as heavy precipitation events where large amount of water is discharged to the lake during a short period. These events are often difficult to track as seepage meters needs to be deployed in this period as well as in the right place."

L296 "because soils are generally wet at this time of year (Sand-Jensen and Lindegaard, 2004)."

a. I recommend saying "saturated" instead of wet.

This has been changed

b. What does that imply? That samples are more representative? But that doesn't go along with the findings that concentrations change during the year.

Yes, this means that the samples are representative. The changes in concentration of CDOM and DOC in the lake are related to the seasonal changes in UV-radiation.

L310 Did you show inter-annual DOC concentrations in the results? If yes refer to Figure, Table. If not provide evidence for the data you refer to.

Data is not shown – we have measured DOC and CDOM seasonally in the groundwater and the lake. Either can be added if the editor wants it?

L346f "Furthermore, huge reductions would occur for TP and TN, with a decrease of 82-96 % if diverting water from the eastern shore in contrast to the southern shore with a modelled decrease of 4-18 %. "What do the 4-18% relate to?

We have added that the 4-18 decrease is TP and TN as well.

"Furthermore, huge reductions would occur for TP and TN, with a decrease of 82-96 % if diverting water from the eastern shore in contrast to the southern shore with a modelled decrease of 4-18 % in TP and TN."

**Technical comments**

All technical comments have been corrected

L17 ... was estimated to be 2 years.

L17 ...Isolation of groundwater recharge areas was based on...

L18 ... ...sites with a high degree... were isolated... (I would also consider saying "identified" instead of "isolated")

L36 I recommend substituting "changes" by "differences"

L37 ... particularly in small water...

L19 I recommend exchanging "Although" by "However"  L25 I recommend exchanging "in relation to lake water..." by "comparing to lake water..." L30 "... groundwater contributes nutrients...."  skip "with"

L44 "... which quantify..."

L46 Exchange "although" by "However"

L50 "...to determine the groundwater input and influence."

L59 "the fate IS well known"

L60 "...which are either remineralized..."

L98 "...hermetically closed..."

L119 "...corrected for the inner filter effect."

L122 "This allows for the detection of components insufficiently represented...."

L123 Reference to software (probably R) is missing

L124ff "A contour map showing the measured FDOM concentrations in groundwater was plotted in ArcMap

__

(ArcMap 10.4.1, ESRI, U.S.A) using the inverse distance weighted (IDW) function ...."

L144f "... the average trait value of all species..."
L175ff "Thus, enabling estimations of the specific removal on a monthly basis related to the concentration measured in the lake at the sampling time following Eq. (4)"This is not a complete sentence. Please add a verb.

L192 "Component C1 was similar to previously found humic-like material..."

 L192ff "The component absorbs in the UV-C region which has low intensities at the ground surface (Diffey, 2002) and are is therefore expected to be photo-resistant (Ishii and Boyer, 2012)."

L197f  "The component absorbs in the UV-A region and are is susceptible to both microbial and photochemical degradation..."

L208   "...with a lowest value of only 0.1 R.U...."

L209  "Components C1, C2, and C3 had..."

L210  "Concentrations of C4 were generally higher...."

L253 "While 18O worked well as a general groundwater recharge estimator, it does not indicate which sites deliver more water." about the last part of the sentence: Consider rephrasing including "quantitative information"

After this, I have given up to comment on each of the technical errors.

---

## Author Response (AR1)

*Comments to the Author:*

*Dear authors,*

*After the very detailed reviews and based on your responses I decided to accept your manuscript for publication with the indicated changes. As both reviewers were concerned about gramar/language, I suggest to perform an additional proofreading through a native speaker and/or professional in the final version.*

*Sincerely,*

*Florian Wittmann*

Dear Florian

We have incorporated the indicated changes and had the manuscript proofread by a professional. Thus, resulting in some minor sentence changes as well as overall corrections.

Best regards Emil Kristensen